# Cell non-autonomous functions of S100a4 drive fibrotic tendon healing

Jessica E Ackerman, Anne EC Nichols, Valentina Studentsova, Katherine T Best, Emma Knapp, Alayna E Loiselle*

Center for Musculoskeletal Research, Department of Orthopaedics and Rehabilitation, University of Rochester Medical Center, Rochester, United States

**Abstract** Identification of pro-regenerative approaches to improve tendon healing is critically important as the fibrotic healing response impairs physical function. In the present study we tested the hypothesis that S100a4 haploinsufficiency or inhibition of S100a4 signaling improves tendon function following acute injury and surgical repair in a murine model. We demonstrate that S100a4 drives fibrotic tendon healing primarily through a cell non-autonomous process, with S100a4 haploinsufficiency promoting regenerative tendon healing. Moreover, inhibition of S100a4 signaling via antagonism of its putative receptor, RAGE, also decreases scar formation. Mechanistically, S100a4 haploinsufficiency decreases myofibroblast and macrophage content at the site of injury, with both cell populations being key drivers of fibrotic progression. Moreover, S100a4-lineage cells become $\alpha$-SMA$^+$ myofibroblasts, via loss of S100a4 expression. Using a combination of genetic mouse models, small molecule inhibitors and in vitro studies we have defined S100a4 as a novel, promising therapeutic candidate to improve tendon function after acute injury.

DOI: https://doi.org/10.7554/eLife.45342.001

## Introduction

Tendons are composed primarily of a dense, highly aligned collagen extracellular matrix (ECM), and connect muscle to bone to transmit mechanical forces throughout the body. Following injury, tendon demonstrates limited regenerative potential and heals through a scar-mediated fibrotic process involving abundant, disorganized ECM deposition. While scar tissue can impart some mechanical strength to the healing tissue, it is also mechanically inferior to native tendon and dramatically impairs normal tendon function resulting in substantial morbidity. In addition, scar tissue increases tendon bulk and forms adhesions to the surrounding tissues, impeding normal range of motion (ROM). This pathological response to injury represents a major clinical burden considering there are over 300,000 surgical tendon repairs in the United States annually (*Pennisi, 2002*), and a high proportion of primary tendon repairs heal with unsatisfactory outcomes and impaired function (*Aydin et al., 2004*; *Galatz et al., 2004*). Despite this burden, there is currently no consensus biological or pharmacological approach to improve tendon healing, due in large part to a paucity of information on the cellular and molecular components involved.

S100a4 (also known as *Fsp1*, *Mts1*, *Pk9a*) is a member of the S100 family of EF-hand Ca$^{2+}$-binding proteins, and is a potent regulator of fibrosis in many tissues including the liver (*Chen et al., 2015*; *Louka and Ramzy, 2016*), lung (*Lawson et al., 2005*), heart (*Tamaki et al., 2013*) and oral submucosa (*Yu et al., 2013*). An increase in the proportion of S100a4$^+$ cells is characteristic of many fibrotic conditions (*Flier et al., 2010*; *Lawson et al., 2005*), and elevated serum S100a4 levels positively correlate with fibrosis clinically (*Chen et al., 2015*). Moreover, the therapeutic potential of S100a4 inhibition is suggested by S100a4-cell depletion studies and S100a4 RNAi treatments in which fibrosis was halted, or effectively reversed (*Chen et al., 2015*; *Iwano et al., 2001*; *Okada et al., 2003*). While depletion of S100a4$^+$ cells can inhibit fibrotic progression, S100a4 can

*For correspondence:
alayna_loiselle@urmc.rochester.
edu

Competing interests: The authors declare that no competing interests exist.

also function as an intra- and extracellular signaling molecule to impact cellular processes including motility, survival, differentiation, and contractility (*Björk et al., 2013*; *Schneider et al., 2008*). Additionally, the effects of S100a4 are cell and tissue-type dependent. We have previously shown that *S100a4*-Cre efficiently targets resident tendon cells (*Ackerman et al., 2017*), however, the specific function of S100a4 and whether that function is primarily cell-autonomous or cell non-autonomous during scar-mediated fibrotic tendon healing is unknown.

In the present study we delineate the relative contributions of S100a4 expression and S100a4$^+$ cells to scar-mediated tendon healing and investigated the effects of S100a4 haploinsufficiency on macrophage and tendon cell function, the cell non-autonomous extracellular signaling function of S100a4 during tendon healing, as well as the fate and function of S100a4-lineage cells in the healing tendon. We have identified S100a4 haploinsufficiency as a novel model of regenerative tendon healing and defined a requirement for S100a4$^+$ cells in the restoration of mechanical properties during tendon healing. These data identify S100a4 signaling as a novel target to improve tendon healing and demonstrate the efficacy of pharmacological inhibition of S100a4 signaling to improve functional outcomes during healing.

## Results

### S100a4 is expressed by resident tendon cells and the S100a4$^+$ population expands during healing

Spatial localization of S100a4 was examined before and after flexor digitorum longus (FDL) tendon repair surgery in S100a4-Cre; ROSA-Ai9 reporter mice to trace S100a4-lineage cells (S100a4$^{Lin+}$; *Figure 1A & B*), and S100a4-GFP$^{promoter}$ mice to identify cells actively expressing S100a4 (S100a4-GFP$^{promoter+}$; *Figure 1D and E*). Most resident tendon cells were S100a4$^{Lin+}$ in the uninjured tendon. Following tendon repair, S100a4$^{Lin+}$ cells were located in the native tendon and bridging scar tissue at D7 and D14 post-surgery (*Figure 1B*). Quantitatively, there was a transient reduction in the S100a4$^{Lin+}$ area at D7, relative to un-injured tendon. However, by 14 there was no significant difference (*Figure 1C*). Many resident tendon cells were actively expressing S100a4 at baseline, however many S100a4$^-$ cells were also present (*Figure 1E*). Following injury, there were abundant S100a4-GFP$^{promoter+}$ cells in the bridging scar tissue from D3 to D14, with S100a4-GFP$^{promoter+}$ cells persisting at least through D28 (*Figure 1E*). Interestingly, the S100a4-GFP$^{promoter+}$ area was significantly increased at D14 relative to D3, D7 and D28 (p=0.05) (Fig. F). Notably, the persistence of the S100a4-GFP$^{promoter+}$ population was also observed in the healing Achilles tendon (*Figure 1—figure supplement 1*), suggesting potential conservation of S100a4 function between tendons. Consistent with changes in spatial expression over time, *S100a4* mRNA expression increased from D3 to a peak at D10, followed by a progressive decline through D28 (*Figure 1G*).

### S100a4 haploinsufficiency promotes regenerative, mechanically superior tendon healing

To determine the functional implications of decreasing *S100a4* expression during FDL tendon healing (*Figure 2A*), we utilized S100a4 haploinsufficient mice (S100a4$^{GFP/+}$), which results in a 50% reduction in *S100a4* mRNA expression in the tendon (*Figure 2B*), as well as a robust decrease in S100a4 protein expression during tendon healing (*Figure 2C*). S100a4 haploinsufficiency did not alter baseline tendon function, with no significant differences observed in MTP Flexion Angle (p=0.22), Gliding Resistance (p=0.094), max load at failure (p=0.4), or stiffness (p=0.6) in un-injured contralateral control tendons (*Figure 2—figure supplement 1*). In addition, decreased *S100a4* expression did not noticeably alter the spatial localization of S100a4$^+$ cells in either the un-injured tendon or at D14 post-surgery (*Figure 2—figure supplement 2*). However, at D14 post-surgery, functional outcomes of scar formation in healing S100a4$^{GFP/+}$ tendons were significantly improved compared to WT. A significant 36% increase in MTP Flexion Angle was observed in S100a4$^{GFP/+}$ repairs, relative to WT (p=0.04) (*Figure 2D*). Gliding Resistance was significantly decreased by 43% in S100a4$^{GFP/+}$ repairs, relative to WT (p=0.028) (*Figure 2E*), suggesting a reduction in scar formation in S100a4$^{GFP/+}$ repairs. In addition, maximum load at failure was significantly increased (+35%) in S100a4$^{GFP/+}$ repairs relative to WT (p=0.003) (*Figure 2F*), while stiffness was increased 28% in S100a4$^{GFP/+}$ repairs, relative to WT, however this increase was not statistically significant (p=0.08)

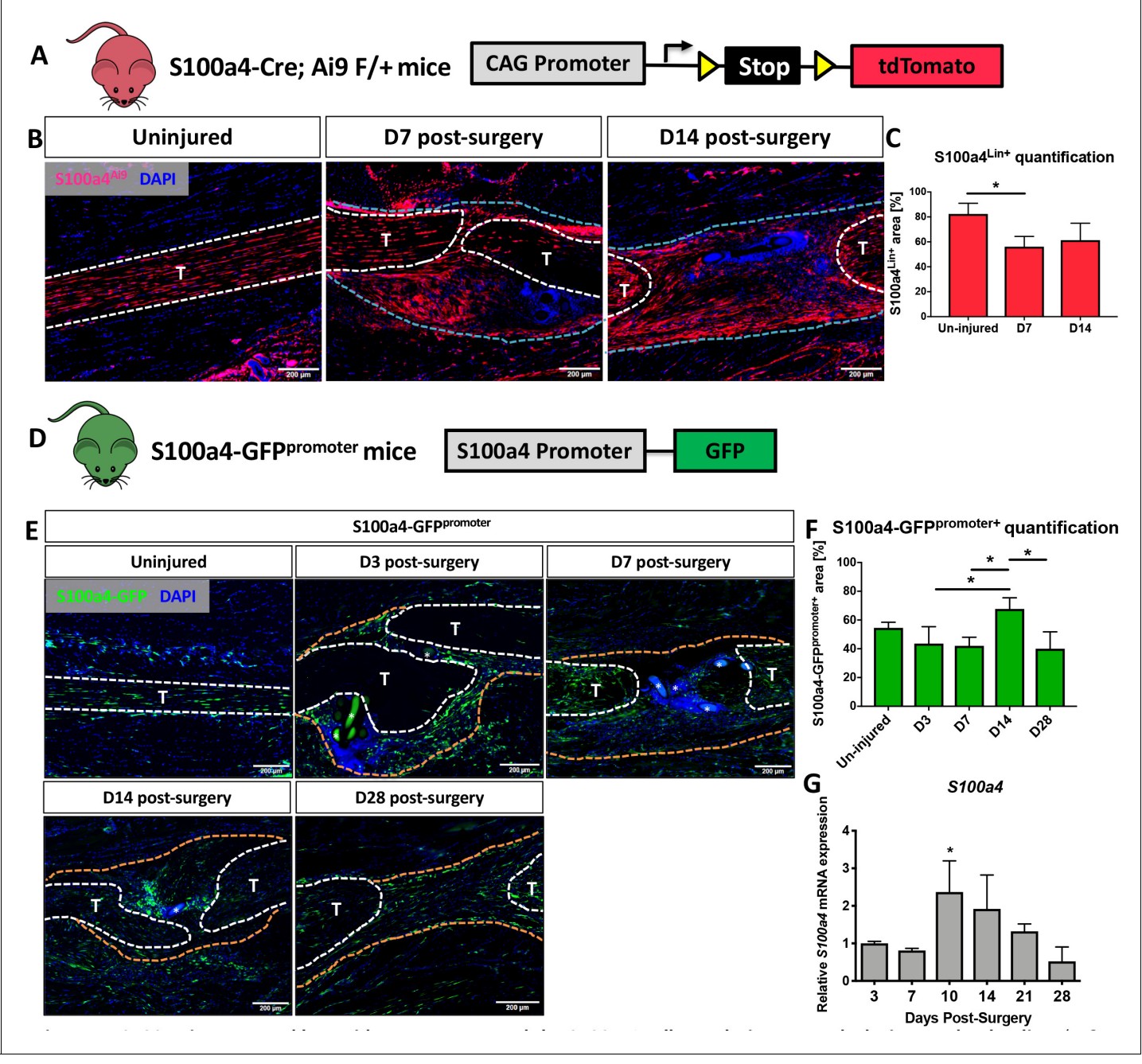

**Figure 1.** S100a4 is expressed by resident tenocytes and the S100a4+cell population expands during tendon healing. (**A and B**) S100a4-Cre; Rosa-Ai9 reporter mice demonstrate efficient targeting of resident tendon cells. Following injury, the S100a4-lineage (S100a4Lin+) population expands, with S100a4Lin+ cells in the native tendon stubs and the bridging scar tissue at D7 and D14 post-surgery. Tendons are outlined in white, and bridging granulation tissue outlined in blue. (**C**) Quantification of S100a4Lin+ area over time. (*) indicates p<0.05 (1-way ANOVA). (**D**) The S100a4-GFPpromoter construct identifies cells actively expressing S100a4 (S100a4-GFPpromoter+). (**E**) A subpopulation of resident tenocytes is S100a4-GFPpromoter+ at baseline, and the S100a4-GFPpromoter+ population increases following injury, with S100a4-GFPpromoter+ cells observed in the bridging scar tissue and native tendon ends through D28 post-surgery. Tendons are outlined in white, and bridging granulation tissue outlined in orange, (*) identifies sutures. (**F**) Quantification of the S100a4-GFPpromoter+ area over time. (*) indicates p<0.05 (1-way ANOVA). (**G**) qPCR analysis of S100a4 during tendon healing demonstrates peak *S100a4* expression at D10, followed by a progressive decline through D28 (n = 3 per time-point). (*) indicates p<0.05 vs. D3 repair (1-way ANOVA). Data were normalized to expression in D3 repairs, and the internal control β-actin.

DOI: https://doi.org/10.7554/eLife.45342.002

The following figure supplement is available for figure 1:

**Figure supplement 1.** S100a4+cells are found in the healthy and healing Achilles tendon.

*Figure 1 continued on next page*

*Figure 1 continued*

DOI: https://doi.org/10.7554/eLife.45342.003

(*Figure 2G*). Taken together, these data suggest that S100a4 haploinsufficiency improves functional outcomes, while also improving tendon strength.

### S100a4 haploinsufficiency improves tendon morphology and decreases myofibroblast content

Morphologically, both Alcian blue Hematoxylin/Orange G (ABH/OG) and picrosirius red staining demonstrate collagen fibers bridging the tendon ends in both WT and S100a4 haploinsufficient mice at D14 (blue arrows, *Figure 3A*). Quantitatively, *Col1a1* expression was significantly increased 5.6-

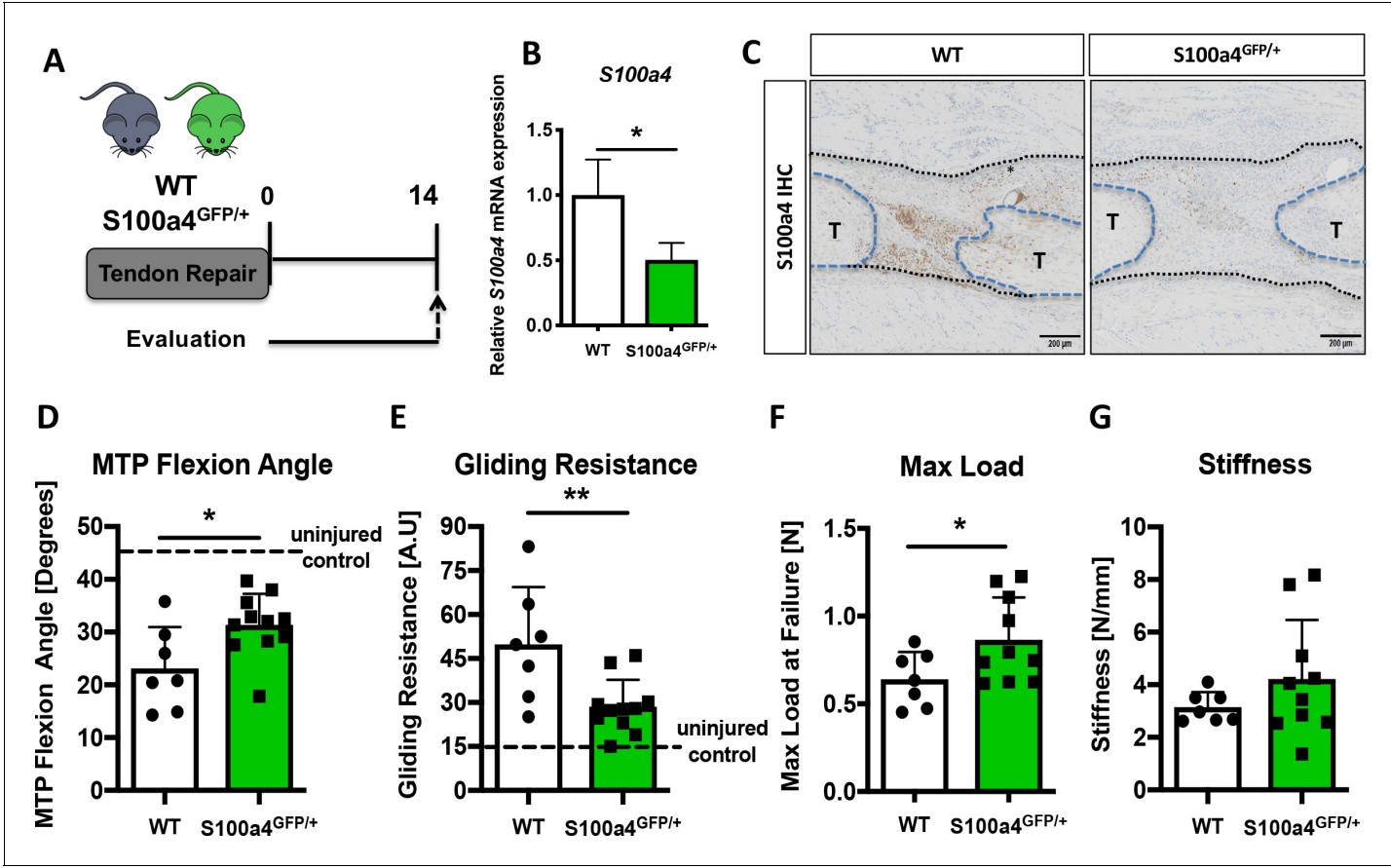

**Figure 2.** S100a4 haploinsufficiency promotes regenerative, mechanically superior tendon healing. (**A**) S100a4GFP/+ haploinsufficient and wild type (WT) littermates underwent transection and repair of the FDL tendon, and tendons were harvested at D14 post-surgery. (**B**) *S100a4* mRNA expression was reduced by 50% in S100a4GFP/+ tendon repairs, relative to WT (n = 3 per group). (**C**) A substantial reduction in S100a4 protein expression was observed in S100a4GFP/+ tendon repairs, relative to WT. Tendon ends are outlined in blue and bridging scar tissue outlined in black (n = 3–4 per group). (**D–G**) At D14, MTP Flexion Angle was significantly increased in S100a4GFP/+ repairs (**D**), and Gliding Resistance was significantly decreased in S100a4GFP/+ repairs (**E**). Max load at failure was significantly improved in S100a4GFP/+ repairs (**F**), while no change in Stiffness was observed between genotypes (**G**) (n = 7–10 per group). (*) indicates p<0.05, (**) indicates p<0.01 between genotypes, n = 7–10 for (**D–G**) (un-paired t-test).

DOI: https://doi.org/10.7554/eLife.45342.004

The following figure supplements are available for figure 2:

**Figure supplement 1.** S100a4 haploinsufficiency does not alter gliding function or mechanical properties of un-injured tendons.
DOI: https://doi.org/10.7554/eLife.45342.005
**Figure supplement 2.** S100a4GFP/+ mice permit tracing of S100a4 haploinsufficient cells.
DOI: https://doi.org/10.7554/eLife.45342.006

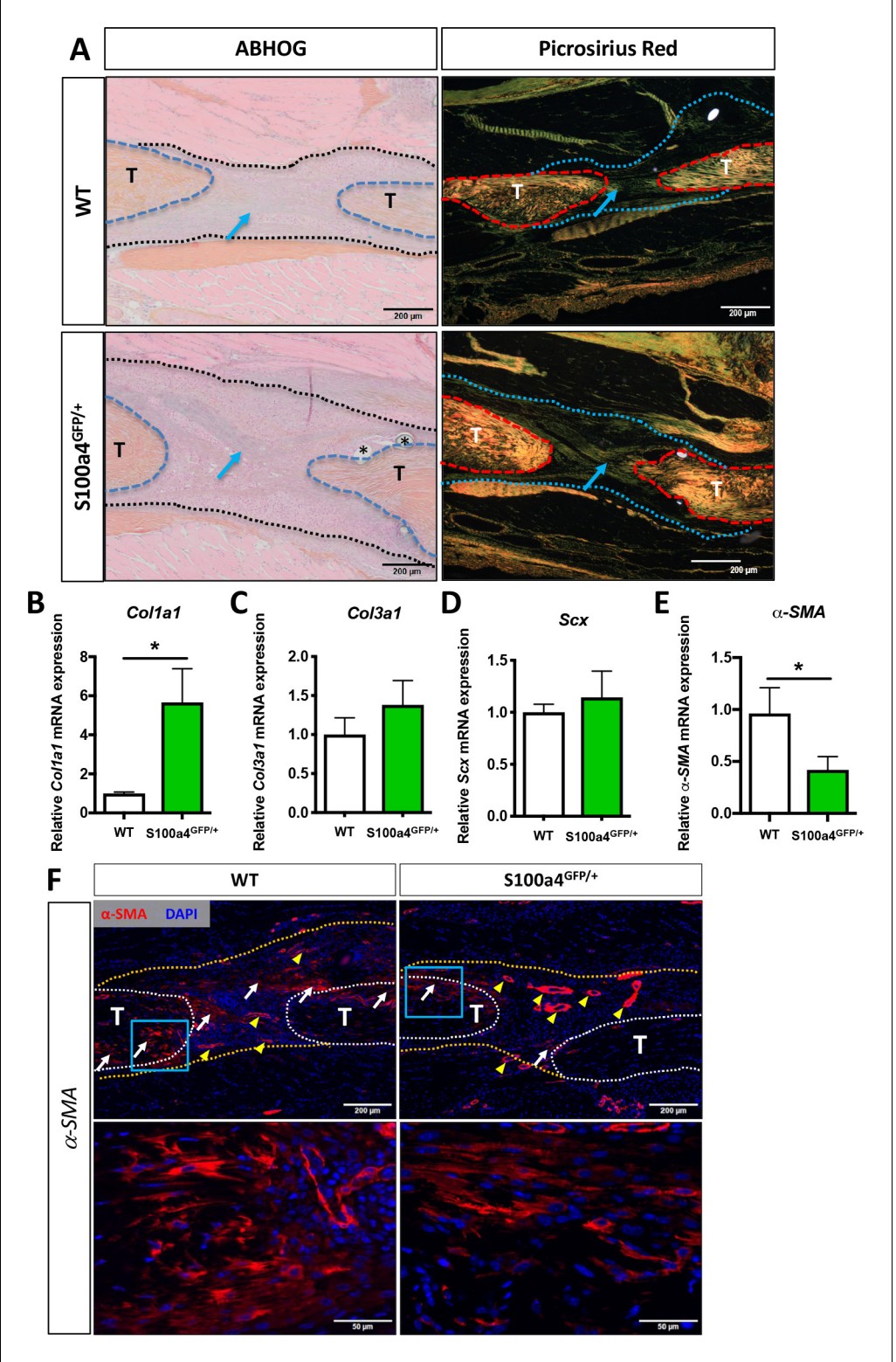

**Figure 3.** S100a4 haploinsufficiency enhances deposition of a mature Collagen matrix and reduced myofibroblast content. (**A**) ABH/OG and picrosirius red staining demonstrate an increase in mature collagen fibers (blue arrows) bridging the tendon ends in S100a4$^{GFP/+}$ repairs compared to WT littermates (n = 3–4 per group) (*) indicate sutures. (**B–D**) S100A4$^{GFP/+}$ tendons expressed significantly more *Col1a1* mRNA (**B**), while transcript levels of *Col3a1* (**C**) and *Scx* (**D**) were unaffected by S100a4 haploinsufficiency. (*) indicates p<0.05 (un-paired t-test), n = 3 per group. (**E** and **F**) *α-SMA* mRNA expression was significantly decreased in S100a4$^{GFP/+}$ repairs (**E**) (n = 3 per

*Figure 3 continued on next page*

*Figure 3 continued*
group), while a substantial reduction in α-SMA protein expression was observed in S100a4$^{GFP/+}$, relative to WT, using immunofluorescence (F). White arrows indicate areas of α-SMA$^+$ cells in the healing tissue, yellow arrowheads denote α-SMA staining of vessels, blue boxes indicate location of higher magnification images.
DOI: https://doi.org/10.7554/eLife.45342.007

fold (p=0.0095) in S100a4$^{GFP/+}$ repairs, relative to WT repairs (*Figure 3B*), while no change in *Col3a1* or *Scx* were observed between groups at D14 (*Figure 3C and D*). Expression of the myofibroblast marker *α-SMA* was decreased 2.4-fold in S100a4$^{GFP/+}$ repairs, relative to WT (p=0.02) (*Figure 3E*). Consistent with this, a marked decrease in α-SMA staining was also observed in S100a4$^{GFP/+}$ repairs, relative to WT (white arrows, *Figure 3F*). These data suggest that S100a4 haploinsufficiency promotes regenerative tendon healing via deposition of a Col1 ECM and a decrease in pro-fibrotic myofibroblasts.

## S100a4 modulates macrophage content and function

Given the fibrotic nature of scar-mediated tendon healing, and the ability of macrophages to modulate multiple aspects of the fibrotic process (*Gibbons et al., 2011*; *Murray et al., 2011*; *Wynn and Ramalingam, 2012*; *Wynn and Vannella, 2016*), we examined changes in macrophage content and polarization during the proliferative phase of healing in WT and S100a4$^{GFP/+}$ mice. While an apparent reduction in F4/80$^+$ macrophages was observed in S100a4$^{GFP/+}$ repairs, relative to WT at D14 (white arrows, *Figure 4A*), no differences in the F4/80$^+$ percent area were observed between genotypes (*Figure 4D*), likely due to the reduced area of scar tissue. In terms of polarization, fewer iNOS$^+$ (M1 marker) macrophages were observed in S100a4$^{GFP/+}$ repairs at D14 (*Figure 4B*), while a trending decrease in the iNOS$^+$ percent area was also observed (p=0.08) (*Figure 4D*). Expression of the M2 macrophage marker IL1ra was not different between WT and S100a4$^{GFP/+}$ at D14 (*Figure 4C*), and no difference in IL1ra$^+$ percent area was observed (*Figure 4D*). Taken together, these data suggest that S100a4 haploinsufficiency may suppress macrophage recruitment or retention during tendon healing, and result in a less pro-inflammatory macrophage environment.

To begin to define the S100a4$^+$ cell population(s) during healing, and to determine if macrophages express S100a4 during tendon healing, we labeled macrophages (Csf1r$^{Lin+}$) and assessed overlapping expression of S100a4 (S1004-GFP$^{promoter+}$). At D3 post-surgery there were many Csf1r$^{Lin+}$; S100a4GFP$^{promoter+}$ cells (white arrows, *Figure 4—figure supplement 1*), however there was a large proportion of cells that were only Csf1r$^{Lin+}$ or S100a4-GFP$^{promoter+}$. By D14 a few Csf1r$^{Lin+}$; S100a4GFP$^{promoter+}$ cells were observed (white arrows, *Figure 4—figure supplement 1*), however, this population was markedly reduced relative to D3. Taken together these data suggest that additional populations of cells express S100a4 during tendon healing, and that the predominant role for S100a4$^+$ macrophages may be during the early phases of healing.

To determine the effects of exogenous S100a4 and S100a4 haploinsufficiency on macrophages in vitro, primary bone marrow derived macrophages (BMDMs) were isolated from C57BL/6J, S100a4$^{GFP/+}$ and WT mice. S100a4 recombinant protein (S100a4-RP) treatment enhanced C57BL/6J BMDM migration, relative to vehicle-treated cells, with a significant increase at the highest dose of 1000 ng/mL S100a4-RP (*Figure 4—figure supplement 2A*). No changes in macrophage migration were observed between vehicle-treated WT and S100a4$^{GFP/+}$ macrophages. In addition, treatment with 50 ng/mL and 1000 ng/mL S100a4-RP enhanced macrophage migration in both WT and S100a4$^{GFP/+}$ cells, relative to vehicle-treated cells (*Figure 4—figure supplement 2B*), suggesting that S100a4 haploinsufficiency in macrophages does not alter migration ability, or responsiveness to S100a4, and indicating that S100a4 modulates macrophage migration via cell non-autonomous effects.

## S100a4-RP treatment of primary macrophages had a variable impact on polarization

Expression of the M1 markers *iNOS* and *CD64* were significantly up-regulated in a dose-dependent manner following S100a4-RP treatment of C57BL/6J BMDMs, while *TNFα* was downregulated and no change was observed in *CD86* expression (*Figure 4—figure supplement 2C*). M2 markers *Arg1*

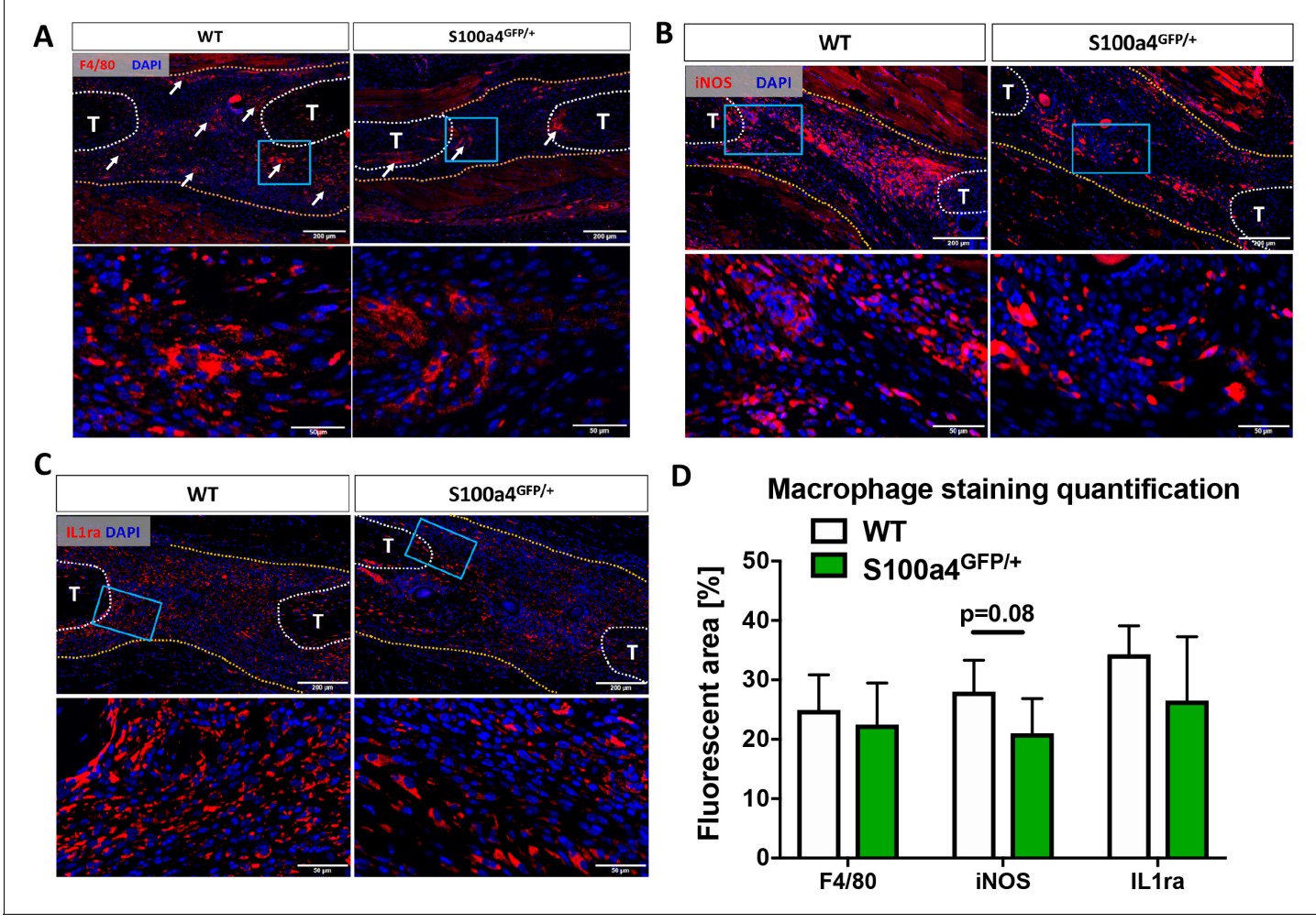

**Figure 4.** S100a4 haploinsufficiency alters the macrophage response to tendon injury. (**A**) F4/80 staining demonstrates decreased macrophage content in the healing tendon of S100a4GFP/+ repairs at D14. White arrows identify concentrated areas of macrophages. (**B**) Expression of the M1 macrophage marker iNOS is markedly reduced in S100a4GFP/+ repairs at D14. (**C**) Expression of the M2 macrophage marker IL1ra is not different between WT and S100a4GFP/+ repairs at D14. Tendon ends are outlined in white, scar tissue is outlined in yellow, blue boxes indicate location of higher magnification images (n = 4 per group). (**D**) The percent area of F4/80+, iNOS+ and IL1ra+ staining, normalized to tissue area was quantified (n = 4) (un-paired t-test).

DOI: https://doi.org/10.7554/eLife.45342.008

The following figure supplements are available for figure 4:

**Figure supplement 1.** S100a4 is expressed by macrophages during early tendon healing.

DOI: https://doi.org/10.7554/eLife.45342.009

**Figure supplement 2.** S100a4 promotes macrophage migration and alters polarization.

DOI: https://doi.org/10.7554/eLife.45342.010

**Figure supplement 3.** Tendon cell S100a4 haploinsufficiency does not alter tenogenic and matrix gene expression or proliferation but enhances migration.

DOI: https://doi.org/10.7554/eLife.45342.011

and *IL1ra* were significantly up-regulated with higher doses of S100a4-RP, while *CD163* was down-regulated and no change in *CD206* expression was observed in C57BL/6J BMDMs (*Figure 4—figure supplement 2D*). No differences in macrophage polarization were observed in vehicle-treated WT and S100a4GFP/+ BMDMs, and in contrast to the C57BL/6J BMDMs, S100a4-RP (1000 ng/mL) treatment had minimal effects on M1 or M2 polarization (*Figure 4—figure supplement 2E and F*).

Given that not all S100a4-GFPpromoter+ cells are Csf1rLin+ during healing and that resident tendon cells express S100a4 (*Figure 1E*), we also examined the cell autonomous effects of S100a4 haploinsufficiency in primary tendon cells. A significant 50% reduction in *S100a4* expression was confirmed

in S100a4$^{GFP/+}$ tenocytes, while no significant differences in tenogenic marker expression (*Scx*, *Tnmd*, *Mkx*) or matrix gene expression (*Col1a1*, *Col3a1*, *Fn*) were observed between WT and S100a4$^{GFP/+}$ tendon cells (*Figure 4—figure supplement 3A and B*). In addition, no changes in proliferation were observed between WT and S100a4$^{GFP/+}$ tenocytes (*Figure 4—figure supplement 3C*). To assess potential changes in migration we used a scratch wound assay. No changes in wound closure were observed between genotypes between 0–12 hr, however, trending improvements in closure were observed at 8 hr (p=0.059) and 12 hr (p=0.08) in S100a4$^{GFP/+}$ tendon cells. By 24 hr there was a significant increase in percent wound closure in S100a4$^{GFP/+}$ tendon cells, relative to WT (p=0.017) (*Figure 4—figure supplement 3D*), suggesting that S100a4 may modulate tendon cell migration through a cell autonomous process.

## Inhibition of S100a4 signaling, via antagonism of RAGE improves tendon healing

Considering that S100a4 haploinsufficiency improves tendon healing, and S100a4 can function as an extracellular signaling molecule to drive fibrotic progression (*Miranda et al., 2010*; *Tomcik et al., 2015*; *Yammani et al., 2006*), we next examined expression of the putative S100a4 receptor, RAGE (Receptor for Advanced Glycation Endproducts) (*Donato et al., 2013*; *Grotterød et al., 2010*; *Sorci et al., 2013*). RAGE expression was observed throughout the scar tissue during tendon healing, with abundant co-localization of S100a4 and RAGE (*Figure 5A*). Consequently, we investigated the feasibility of inhibiting S100a4 signaling via disruption of S100a4-RAGE interaction using RAGE antagonist peptide (RAP) (*Arumugam et al., 2012*). In vivo, RAP treatment (*Figure 5B*) significantly improved measures of gliding function relative to vehicle treated controls, with a 41% increase in MTP Flexion Angle (p=0.008) (*Figure 5C*), and a 39% decrease in Gliding Resistance (p=0.007) (*Figure 5D*). No differences in maximum load at failure (p=0.57) and stiffness (p=0.30) were observed between groups (*Figure 5E,F*). These data suggest that inhibition of S100a4-RAGE recapitulates the improvements in gliding function seen with S100a4 haploinsufficiency but is insufficient to improve mechanical properties.

## S100a4$^+$ cell ablation results in aberrant matrix deposition during tendon healing

While S100a4 haploinsufficiency and inhibition of S100a4 signaling improves tendon healing, S100a4 can also function in a cell-autonomous manner (*Figure 4—figure supplement 3D*; *Chow et al., 2017*). To determine the effects of S100a4$^+$ cell ablation on tendon healing, we depleted proliferating S100a4$^+$ cells from D5-10 post-surgery (immediately preceding peak *S100a4* expression) using S100a4-thymidine kinase (S100a4-TK) mice (*Figure 6A*). Depletion of S100a4$^+$ cells from D5-10 resulted in a significant 91% reduction in *S100a4* mRNA expression at D10 (p<0.001) (*Figure 6B*), and a substantial reduction in S100a4 protein expression, relative to WT at D14 (*Figure 6C*). Functionally, slight but non-significant improvements in gliding function were observed in S100a4-TK (D5-10), relative to WT (*Figure 6D and E*). However, max load at failure was significantly decreased by 43% (p=0.02) (*Figure 6F*), while stiffness was unchanged (*Figure 6G*). Morphologically, S100a4-TK (D5-10) tendons healed with thinner, more acellular bridging scar tissue between the native tendon ends, compared to the larger, more cellular granulation tissue in WT repairs (*Figure 6H*). Picrosirius staining demonstrated a substantial reduction in bridging ECM in S100a4-TK (D5-10) repairs, relative to WT (*Figure 6I*). In contrast to this, qPCR revealed significant increases in ECM proteins *Col1a1* (3.9-fold, p=0.04) (*Figure 6—figure supplement 1A*) and *Col3a1* (1.9-fold, p=0.033) (*Figure 6—figure supplement 1B*) mRNA expression in S100a4-TK (D5-10) mice. Additionally, significant decreases in the tenogenic transcription factor *Scx* (1.75-fold, p=0.04), and the myofibroblast marker α-SMA (9-fold, p=0.0003) were observed in S100a4-TK (D5-10), relative to WT (*Figure 6—figure supplement 1C & D*). Consistent with this, and the phenotype in S100a4$^{GFP/+}$ repairs, α-SMA staining was markedly reduced in S100a4-TK (D5-10) repairs, relative to WT (*Figure 6—figure supplement 2*), as was total macrophage content (*Figure 6—figure supplement 3*). Taken together, S100a4-cell depletion alters normal matrix deposition during tendon healing, leading to pronounced morphological changes in the scar tissue and a loss of overall strength, while recapitulating the changes in myofibroblast and macrophage populations observed in S100a4$^{GFP/+}$ repairs.

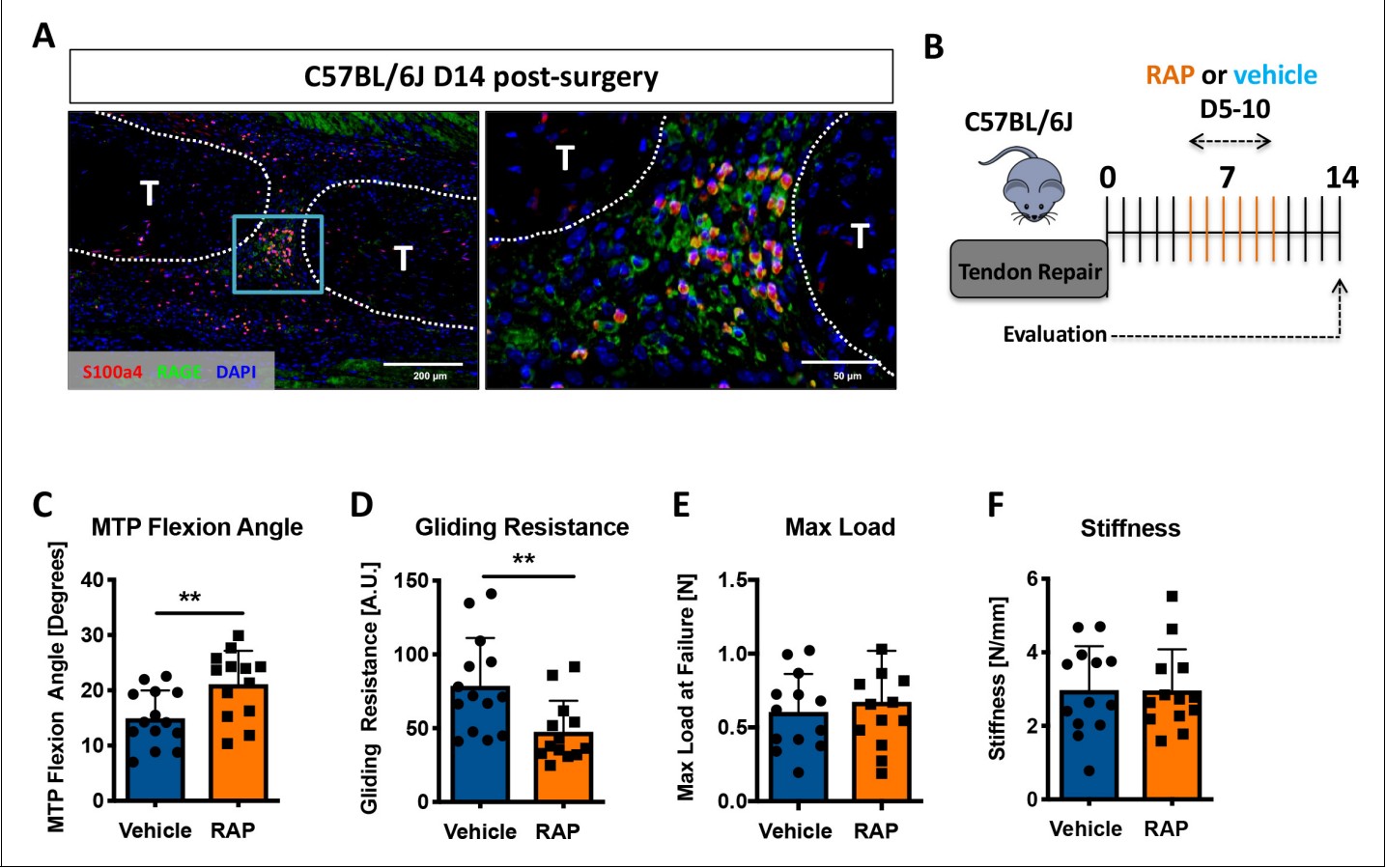

**Figure 5.** Inhibition of S100a4 signaling via RAGE antagonism improves tendon healing. (A) Co-immunofluorescence demonstrated co-localization of S100a4 and its putative receptor RAGE in the healing tendon (n = 3). (B) C57Bl/6J mice were treated with either RAP or vehicle, via i.p. injection from D5-10 post-surgery, and harvested at D14 for functional testing. (C–F) At D14 RAP treatment significantly improved measures of gliding function relative to vehicle, with a (C) significant increase in MTP Flexion Angle, and (D) a significant decrease in Gliding Resistance. No change in (E) Max load at failure, or (F) Stiffness was observed between treatments (n = 13 per group). (**) indicates p<0.01 between treatments (un-paired t-test).
DOI: https://doi.org/10.7554/eLife.45342.012

We then investigated the effects of continuous S100a4[+] cell depletion (D1-14) on healing (*Figure 7A*). In contrast to depletion from D5-10, depletion from D1-14 significantly reduced MTP Flexion Angle (−53%, p=0.0003) (*Figure 7B*), and increased gliding resistance (+187%, p<0.001) (*Figure 7C*), indicating impairment of normal gliding function with sustained S100a4[+] cell depletion. Consistent with D5-10 depletion, depletion from D1-14 reduced mechanical properties, with a 43% decrease in max load (p=0.025) (*Figure 7D*), and a 49% decrease in stiffness (p=0.0078) (*Figure 7E*).

## S100a4-lineage cells represent differentiated a-SMA myofibroblasts in the scar tissue of healing tendon

Examination of S100a4[GFP/+] and S100a4-TK (D5-10) healing tendons demonstrate dramatically reduced myofibroblast content, suggesting potential interplay between S100a4[+] cells and myofibroblasts. The relationship between S100a4 and pro-fibrotic myofibroblasts is controversial and likely tissue-dependent, with conflicting reports of myofibroblast fate for S100a4[+] cells (*Humphreys et al., 2010*; *Li et al., 2018*; *Picard et al., 2008*; *Tanjore et al., 2009*). To understand the relationship between these cell populations during tendon healing we examined α-SMA expression in both S100a4-lineage cells and cells actively expressing S100a4 (S100a4-GFP[promoter+]). S100a4-Cre; Ai9 mice demonstrate that ∼ 65% of α-SMA[+] myofibroblasts at D14 post-surgery are derived from S100a4-lineage, as shown by co-localization (*Figure 8A*, arrows, *Figure 8C*). In contrast, very few (∼16%) S100a4-GFP[promoter+] cells demonstrated co-localization with α-SMA (*Figure 8B and C*).

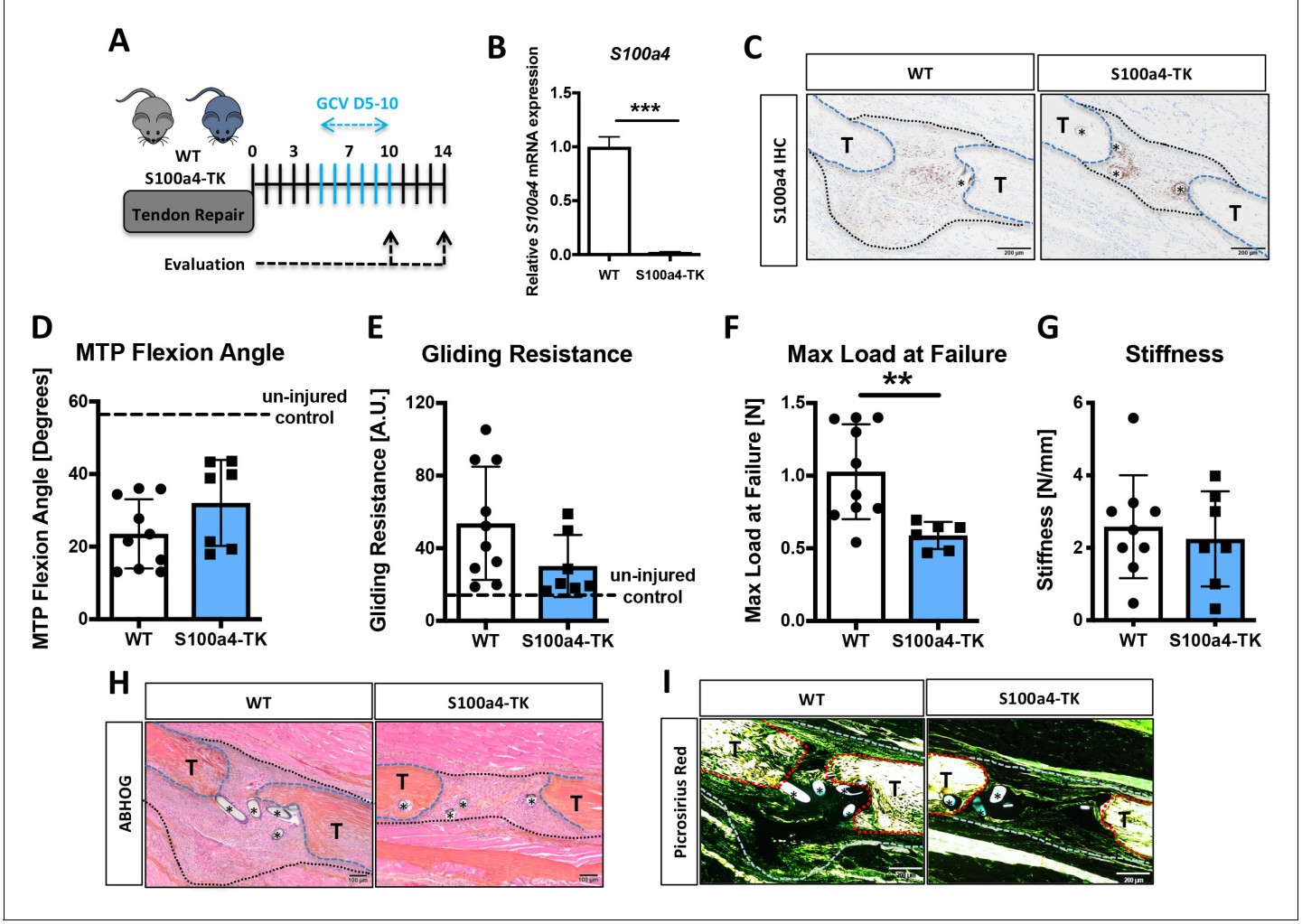

**Figure 6.** Delayed depletion of S100a4+cells impairs restoration of mechanical properties and alters matrix deposition. (**A**) WT and S100a4-TK mice were treated twice daily with ganciclovir (GCV) from D5-10 post-surgery. (**B**) S100a4+ cell depletion results in a 91% reduction in *S100a4* mRNA at D10 post-surgery (n = 3). (**C**) A substantial reduction in S100a4 protein expression was observed S100a4-TK repairs, relative to WT. Tendon is outlined in blue, scar tissue is outlined in black and (*) identify sutures (n = 4). (**D–G**) At D14 no change in MTP Flexion Angle (**D**) and Gliding Resistance (**E**) were observed between WT and S100a4-TK repairs. (**F**) Max load at failure was significantly reduced following S100a4-cell depletion, while no change in Stiffness was observed (**G**) (n = 7–10), (**\*\***) indicates p<0.01 (un-paired t-test). (**H and I**) Morphologically, (**H**) ABH/OG and (**I**) Picrosirius staining demonstrate reduced matrix deposition bridging the tendon ends in the S100a4-TK repairs, relative to WT. (*) Indicates sutures.

DOI: https://doi.org/10.7554/eLife.45342.013

The following figure supplements are available for figure 6:

**Figure supplement 1.** S100a4+cell depletion alters expression of matrix, tenogenic and myofibroblast-associated genes.

DOI: https://doi.org/10.7554/eLife.45342.014

**Figure supplement 2.** S100a4+cell depletion reduces α-SMA+myofibroblast content during healing.

DOI: https://doi.org/10.7554/eLife.45342.015

**Figure supplement 3.** S100a4+cell depletion reduces macrophage content during healing.

DOI: https://doi.org/10.7554/eLife.45342.016

Taken together these data suggest that S100a4-lineage cells lose S100a4 expression during the transition to α-SMA+ myofibroblasts.

## Discussion

Consistent with the role of S100a4 as a key driver of fibrosis (*Bruneval et al., 2005*; *Iwano et al., 2002*; *Lawson et al., 2005*; *Tomcik et al., 2015*), we have demonstrated that S100a4 promotes

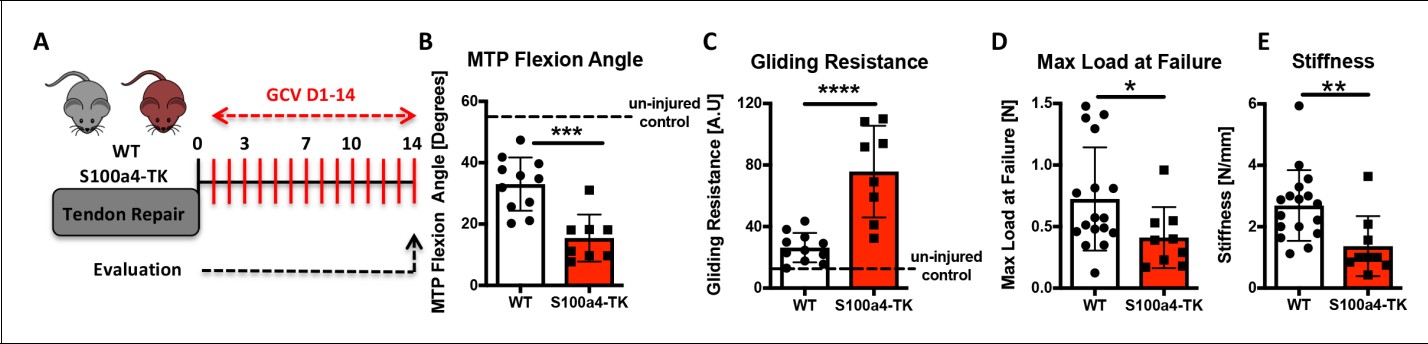

**Figure 7.** Sustained ablation of S100a4$^+$impairs restoration of gliding function and mechanical properties. (A) WT and S100a4-TK mice were treated with GCV from D1-14 post-surgery to ablate proliferating S100a4$^+$ cells. At D14 (B) MTP Flexion Angle was significantly reduced, and (C) Gliding Resistance was significantly increased in S100a4-TK repairs, relative to WT. (D) A non-significant decrease in Max load at failure and (E) a significant reduction in Stiffness were observed in S100a4-TK repairs (n = 8–11). (*) indicates p<0.05, (**) indicates p<0.01 (un-paired t-test).
DOI: https://doi.org/10.7554/eLife.45342.017

scar-mediated tendon healing primarily via cell-non-autonomous extracellular signaling. Further, we have shown that S100a4 haploinsufficiency drives regenerative tendon healing and depletion of S100a4$^+$ cells disrupts re-acquisition of mechanical properties. Interestingly, the effects of S100a4$^+$ cell depletion on functional metrics were time-dependent, suggesting a period of optimal S100a4 inhibition. Mechanistically, S100a4 haploinsufficiency and S100a4$^+$ cell depletion modulates macrophage content, suggesting the ability of S100a4 to regulate the inflammatory milieu during tendon healing. In addition, S100a4-lineage cells lose expression of S100a4 to become α-SMA$^+$ pro-fibrotic myofibroblasts, which are likely involved in both restoration of matrix integrity and deposition of excess ECM. Taken together, these data establish S100a4 haploinsufficiency as a novel model of regenerative, mechanically superior tendon healing, and identify S100a4 as a potent anti-fibrotic therapeutic candidate to improve tendon healing.

S100a4 lacks enzymatic activity and functions predominantly through the regulation and interaction with other proteins. While the intracellular functions of S100a4 are not well-characterized, the extracellular signaling functions of S100a4 include regulation of multiple cellular processes important in fibrosis including motility (*Belot et al., 2002*; *Schmidt-Hansen et al., 2004*), differentiation (*Novitskaya et al., 2000*; *Schneider et al., 2007*; *Stary et al., 2006*) and survival (*Schneider et al., 2007*). S100a4 protein levels are strongly correlated with idiopathic pulmonary fibrosis (*Li et al., 2018*), and S100a4 has been suggested as a potential fibrotic biomarker in the liver (*Chen et al., 2015*). Consistent with this, we see highest S100a4 expression immediately prior to the period of peak scar formation during tendon healing. The therapeutic potential of targeting S100a4 has been well established in the lung, with decreased fibrotic progression following treatment with an S100a4 neutralizing antibody (*Li et al., 2018*), and pharmacological inhibition of S100a4 (*Zhang et al., 2018*). Moreover, *Tomcik et al. (2015)* demonstrated that deletion of S100a4 prevented bleomycin-induced skin fibrosis. Consistent with these studies, we demonstrate that S100a4 haploinsufficiency is sufficient to attenuate scar-mediated tendon healing and promotes a more regenerative healing response. More specifically, S100a4$^{GFP/+}$ repairs demonstrate suppression of multiple components of the fibrotic cascade including macrophage content, myofibroblasts, as well as an altered ECM balance toward a mature tendon composition (*Col1a1 > Col3a1*).

Macrophages play an essential role in wound healing, and S100a4 is a potent chemokine and regulator of macrophage chemotaxis. Bone marrow-derived macrophages from S100a4$^{-/-}$ mice display defects in chemotactic motility and impaired recruitment to the site of inflammation (*Li et al., 2010*). In contrast, we show that S100a4 haploinsufficient primary BMDMs do not exhibit deficits in migration relative to WT, and that S100a4$^{GFP/+}$ cells are responsive to the pro-migration effects of exogenous S100a4. Importantly, it is unknown whether S100a4$^{-/-}$ macrophages also increase migration in response to exogenous S100a4. Answering this question will help clarify whether the cell non-autonomous effects of S100a4 on macrophages are dependent on a minimum level of S100a4 expression in macrophages. Taken together, the enhanced migration of BMDMs in response to S100a4, and the

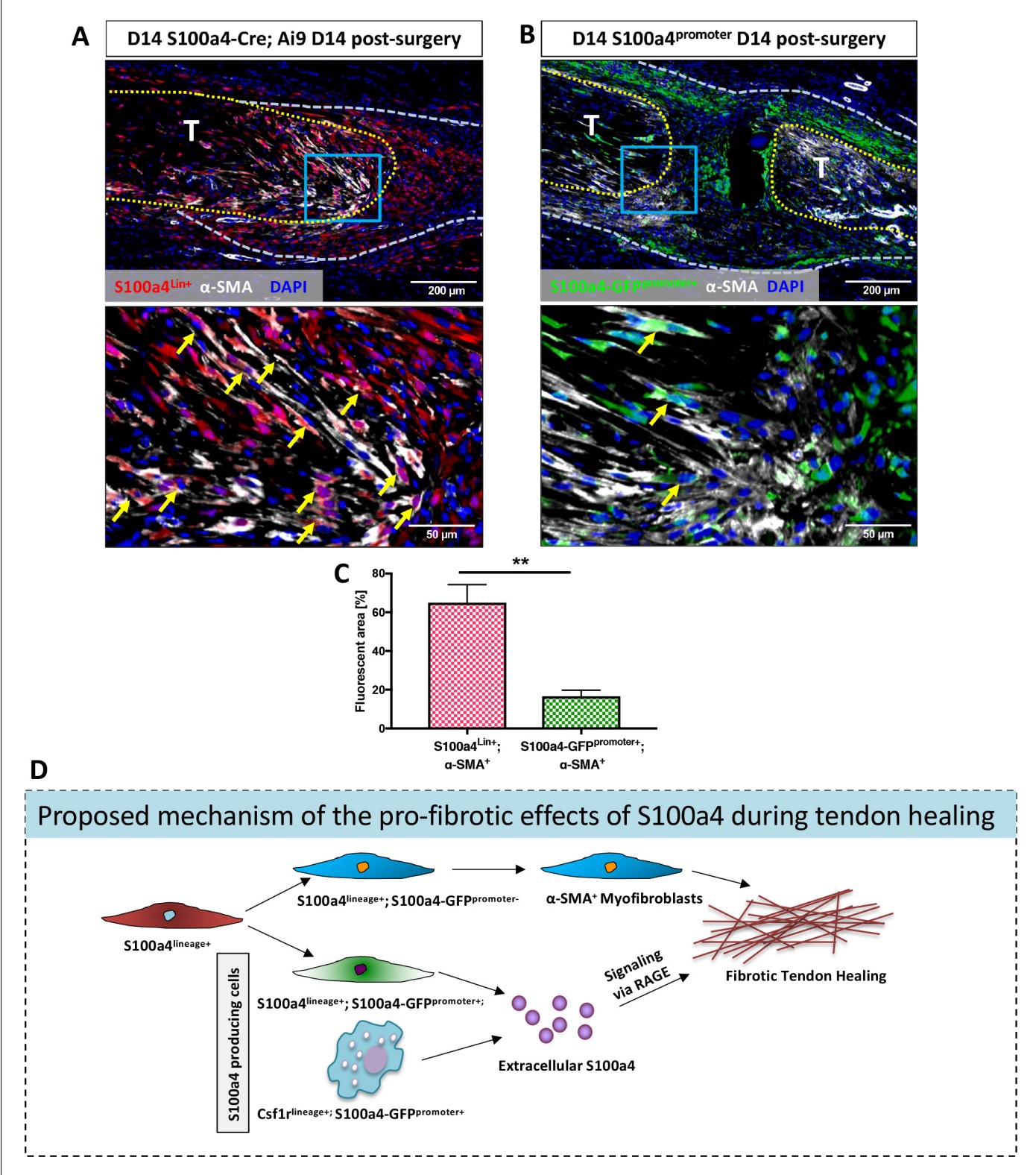

**Figure 8.** S100a4-lineage cells lose S100a4 expression during the transition to α-SMA⁺ myofibroblasts. (**A**) Co-localization of Red Fluorescent Protein (S100a4$^{Lin+}$ cells; red) and the myofibroblast marker α-SMA (white) demonstrated abundant co-localization (yellow arrows) during tendon healing. (**B**) Minimal co-localization of α-SMA (white) and cells actively expressing S100a4 (S100a4-GFP$^{promoter+}$; green) was observed during healing (n = 3). (**C**) Quantification of the percent α-SMA⁺ area that is also S100a4$^{Lin+}$ (red and white bar) or S100a4-GFP$^{promoter+}$ (green and white bar) at D14 (n = 3–4 per group) (**) indicates p<0.01 between groups (un-paired t-test). (**D**) Schematic representation of the proposed cell non-autonomous signaling functions

*Figure 8 continued on next page*

*Figure 8 continued*

of S100a4 in fibrotic healing, as well as cell fate of S100a4-lineage cells. The identities and discrete functions of specific populations of S100a4⁺ cells remains to be determined.

DOI: https://doi.org/10.7554/eLife.45342.018

reduced macrophage content in S100a4 haploinsufficient and S100a4⁺ cell-depleted repairs, suggests a potential role for extracellular S100a4 in macrophage recruitment during tendon healing. Moreover, while RAGE can be expressed on a variety of cell types, macrophages have been shown to be the primary source of RAGE in the context of pathology (*Gaens et al., 2014*; *Song et al., 2014*; *Su et al., 2011*; *Sunahori et al., 2006*). Therefore, in addition to decreasing the macrophage content, and thereby RAGE⁺ cells, S100a4^GFP/+ may also suppress signaling via direct down-regulation of RAGE. RAGE expression is highly dependent on ligand concentration (*Bierhaus et al., 2005*), and we have decreased *S100a4* expression by 50%, implying down-regulation of the entire S100a4-RAGE-macrophage axis in S100a4^GFP/+ mice. Interestingly, although inhibition of S100a4-RAGE with RAP recapitulates the improvements in gliding function observed in S100a4^GFP/+ repairs, it is insufficient to improve mechanical properties. This suggests that S100a4 may regulate restoration of mechanical properties through mechanisms independent of RAGE. In addition to RAGE, extracellular S100a4 acts as a ligand for TLR4 (*Björk et al., 2013*.), and has been shown to interact with EGFR ligands (*Klingelhöfer et al., 2009*), Heparan Sulfate Proteoglycans (*Kiryushko et al., 2006*), and cell surface Annexin 2 (*Semov et al., 2005*) to modulate signaling. Understanding the potential RAGE-independent mechanisms of S100a4 signaling will be an important aspect of future research. Importantly, while RAP treatment inhibits S100a4-RAGE signaling, it also blocks binding of other RAGE ligands. In addition, there are likely time-dependent effects of S100a4-RAGE signaling. Since RAP treatment was limited to a single timing regimen it is unknown how inhibition of RAGE signaling at other time-points, or in a sustained fashion may affect healing. Future studies to specifically inhibit S100a4 signaling, and to examine the time-dependent effects of S100a4-RAGE inhibition are needed.

S100a4⁺ cell depletion has shown great efficacy in managing fibrotic progression in the peritoneum (*Okada et al., 2003*), and kidney (*Iwano et al., 2001*). In contrast, depletion of S100a4⁺ cells impairs tendon healing, particularly when S100a4⁺ cells are depleted from D1-14. These opposing effects may be due to fundamental differences in the functions of affected tissues, with tendon being a mechanical, load-bearing tissue. Interestingly, the more pronounced negative effect of cell depletion from D1-14, relative to D5-10 depletion may be explained by suppression of the acute inflammatory phase. Anti-inflammatory administration during the acute inflammatory phase of tendon healing is effective at reducing scar formation, but causes marked reductions in mechanical properties (*Connizzo et al., 2014*; *Dimmen et al., 2009*; *Hammerman et al., 2015*). Taken together, these data further support the importance of timing treatment to modulate fibrotic tendon healing, particularly as it relates to S100a4 inhibition, as noted above.

One of the main controversies surrounding S100a4⁺ cells is their potential to become α-SMA⁺ myofibroblasts. Osterreicher *et al.*, demonstrated that neither actively expressing S100a4⁺ cells or S100a4^Lin+ cells express α-SMA during lung fibrosis (*Österreicher et al., 2011*.), and additional work demonstrates that S100a4⁺ bone marrow cells do not express α-SMA (*Cheng et al., 2012*). In contrast, dermal fibroblasts are positive for expression of both S100a4 and α-SMA (*Österreicher et al., 2011*) and *Chen et al. (2015)* demonstrate that S100a4 treatment increases α-SMA expression, while α-SMA⁺ cells decrease with S100a4-cell depletion in a model of liver fibrosis. Using a similar combination of S100a4-lineage tracing and active S100a4 expression analyses as in *Österreicher et al. (2011)*, we demonstrate that S100a4^Lin+ cells become α-SMA⁺, and that the α-SMA⁺ population is largely negative for S100a4. These data not only define a terminal myofibroblast fate for many S100a4-lineage cells during tendon healing, but further support the concept of exquisite cell and tissue specificity of S100a4 signaling.

In addition to terminal cell fate, the origin of S100a4⁺ cells are unclear. Osterreicher *et al.*, demonstrate that bone marrow derived cells (BMDCs) represent the main source of S100a4 during liver fibrosis (*Österreicher et al., 2011*), and S100a4⁺ BMDCs have been shown to migrate to the site of neointima formation following vein grafting (*Cheng et al., 2012*). While we have not traced S100a4⁺

BMDCs in this study, we have previously shown specific recruitment of BMDCs to the healing tendon (*Loiselle et al., 2012*), and we also see S100a4[Lin+] and S100a4-GFP[promoter+] populations in the tendon without injury. To begin to define the S100a4[+] cells during healing we examined S100a4 expression in Csf1r[Lin+] macrophages as Osterreicher et al., suggest a subpopulation of inflammatory macrophages express S100a4. Somewhat surprisingly, only a small proportion of macrophages express S100a4 at D3 and D14 post-surgery. In addition to macrophages, we also observe S100a4 expression in resident tendon cells prior to injury, however, the specific function of tendon-derived S100a4 is unclear. Taken together, the cellular contributors to S100a4 expression and therefore scar-mediated healing are clearly complex and multi-faceted. Future studies are needed to delineate the relative contributions of S100a4 from different cell populations to the scar-mediated tendon healing process, using cell-type specific, inducible Cre lines.

While we clearly identify S100a4 haploinsufficiency as a model of regenerative tendon healing, there are several limitations, in addition to those discussed above, that must be considered. First, we have not investigated whether this signaling paradigm is conserved in other tendons. However, we do observe S100a4[+] cells in the Achilles tendon at homeostasis and expansion of this population following injury. Since scar-mediated healing is consistent between tendons (*Shepherd et al., 2014*), this reinforces the potential application of this approach to improve healing in multiple tendons. Moreover, the surgical repair and healing model used in these studies does not fully recapitulate the clinical scenario as no rehabilitation is conducted, and there is clear evidence of the beneficial effects of physical therapy to improve healing outcomes (*Starr et al., 2013*). Second, the long-term effects of diminished S100a4 signaling are unknown, as we have only examined healing at D14. Therefore, it will be important to define the long-term effects of S100a4 haploinsufficiency or pharmacological inhibition on the healing process, including assessment of material properties, which were not measured in the current study. However, peak expression of *S100a4* at D10 post-surgery suggests the prime effects of S100a4 may occur during the early inflammatory-proliferative phases of healing. Moreover, the time-dependent effects of S100a4-cell depletion suggest there is likely an optimal therapeutic window for S100a4 inhibition, which will be examined in future studies. Finally, we have not delineated between the effects of S100a4 expression in resident tendon cells relative to expression in extrinsic cells, and how the cell origin of S100a4 may dictate the effects on healing.

Restoring satisfactory function following tendon injury has remained an intractable clinical problem for decades (*Strickland, 2000*). To our knowledge this is the first model of regenerative tendon healing in transgenic mice, defined by improvements in range of motion and mechanics. These data will inform future work to define the pathways down-stream of S100a4-RAGE, rigorously determine the time-dependent effects of S100a4 inhibition, define and delineate the functions of all S100a4[+] cell populations, and identify the cues that drive the transition of S100a4-lineage cells to myofibroblasts. More directly however, these studies define the tremendous potential of inhibition of S100a4 signaling as a therapeutic approach to promote regenerative tendon healing.

# Materials and methods

## Key resources table

| Reagent type (species) or resource | Designation | Source or reference | Identifiers | Additional information |
|---|---|---|---|---|
| Genetic reagent (*Mus. musculus*) | B6.Cg-Tg(S100a4-EGFP) M1Egn/YunkJ (S100A4-GFP[promoter]) | Jackson Laboratory | Stock #: 012893 RRID: MGI:4819362 | |
| Genetic reagent (*M. musculus*) | B6.Cg-Tg(S100a4-TK) M31Egn/YunkJ (S100a4-TK) | Jackson Laboratory | Stock #: 012902 RRID:MGI:4454768 | |
| Genetic reagent (*M. musculus*) | B6.129S6-S100a4tm1Egn /YunkJ (S100a4[GFP/+]) | Jackson Laboratory | Stock #: 012904 RRID:MGI:4819358 | |
| Genetic reagent (*M. musculus*) | BALB/c-Tg(S100a4-cre) 1Egn/YunkJ (S100a4-Cre) | Jackson Laboratory | Stock #: 012641 RRID:MGI:4454332 | |

*Continued on next page*

*Continued*

| Reagent type (species) or resource | Designation | Source or reference | Identifiers | Additional information |
|---|---|---|---|---|
| Genetic reagent (*M. musculus*) | B6.Cg-Gt(ROSA) 26Sortm9(CAG-td Tomato)Hze/J (ROSA-Ai9) | Jackson Laboratory | Stock #: 007909 RRID:MGI:3809523 | |
| Genetic reagent (*M. musculus*) | Tg(Csf1r-Mer-iCre-Mer) 1Jwp (Csf1r-iCre) | Jackson Laboratory | Stock #: 019098 RRID:IMSR_JAX:019098 | |
| Genetic reagent (*M. musculus*) | C57BL/6J | Jackson Laboratory | Stock #: 000664 RRID:MGI:3028467 | |
| Antibody | anti-RAGE (mouse monoclonal) | Santa Cruz Biotechnology | Cat. #: sc-365154 RRID:AB_10707685 | 1:100 |
| Antibody | anti-F4/80 (goat polyclonal) | Santa Cruz Biotechnology | Cat. #: sc-26642 RRID:AB_2098333 | 1:500 |
| Antibody | anti-GFP (goat polyclonal) | Abcam | Cat. #: ab6673 RRID:AB_305643 | 1:5000 |
| Antibody | anti-RFP (rabbit polyclonal) | Abcam | Cat. #: ab62341 RRID:AB_945213 | 1:500 |
| Antibody | anti-S100a4 (rabbit monoclonal) | Abcam | Cat. #: ab197896 RRID:AB_2728774 | 1:20000 |
| Antibody | anti-iNOS (rabbit polyclonal) | Abcam | Cat. #: ab15323 RRID: AB_301857 | 1:100 |
| Antibody | Anti-ILIRa (rabbit monoclonal) | Abcam | Cat. #: ab124962 RRID:AB_11130394 | 1:10000 |
| Antibody | anti-alpha-SMA-Cy3 (mouse monoclonal) | Sigma-Aldrich | Cat. #: C6198 RRID: AB_476856 | 1:250 |
| Antibody | Donkey anti-rabbit AlexaFluor594 secondary | Jackson ImmunoResearch | Cat. #: 711-585-152 RRID: AB_2340621 | 1:200 |
| Antibody | Donkey anti-rabbit 647 secondary | Jackson ImmunoResearch | Cat. #: 711-605-152 RRID: AB_2492288 | 1:200 |
| Antibody | Donkey anti-rabbit Rhodamine-Red-X | Jackson ImmunoResearch | Cat. #: 711-296-152 RRID:AB_2340614 | 1:100 |
| Antibody | Donkey anti-goat 488 secondary | Jackson ImmunoResearch | Cat. #: 705-546-147 RRID: AB_2340430 | 1:200 |
| Antibody | Goat anti-mouse AlexaFluor488 secondary | ThermoFisher | Cat. #: A11029 RRID:AB_138404 | 1:1000 |
| Chemical compound, drug | Nucleoside analog ganciclovir (GCV) | TSZCHEM | Cat #: 82410-32-0 | 75 mg/kg |
| Peptide, recombinant protein | RAGE Antagonist Peptide (RAP) | MilliporeSigma | Cat. #: 553031 | 100 ug (peptide) |
| Peptide, recombinant protein | Human S100a4 Recombinant protein | LSBio | Cat #: G1305 | 20–1000 ng/mL (recombinant protein) |
| Commercial kit | Rabbit polymer kit | Vector Laboratories | Cat #: MP-7401 | |
| Software | OlyVIA software | Olympus (https://www.olympus-lifescience.com/en/support/downloads/) | RRID:SCR_016167 | Version 2.9 |
| Software | ImageJ software | ImageJ (http://imagej.nih.gov/ij/) | RRID:SCR_003070 | |
| Software | GraphPad Prism software | GraphPad Prism (https://graphpad.com) | RRID:SCR_015807 | Version 8.0.0 |

### Ethics statement

All animal studies were approved by the University of Rochester Committee for Animal Resources.

### Mouse strains

S100A4-GFP$^{promoter}$ mice (#012893), S100a4-TK (#012902), S100a4$^{GFP/+}$ (#012904), S100a4-Cre (#012641), ROSA-Ai9 (#007909), Csf1r-iCre (#019098), and C57BL/6J (#000664) were acquired from The Jackson Laboratory (Bar Harbor, ME).

S100a4-GFP promoter mice contain a construct encoding EGFP under control of the S100a4 promoter sequence, resulting in green fluorescence in cells actively expressing S100a4 (*Iwano et al., 2002*). S100a4-TK mice contain a viral thymidine kinase gene downstream of the S100a4 promoter, and treatment with the nucleoside analog ganciclovir (GCV) halts DNA replication in proliferating cells expressing S100a4, resulting in apoptosis and S100a4$^+$ cell ablation (*Iwano et al., 2001*). Mice were treated twice per day (i.p) with 75 mg/kg GCV. S100a4$^{GFP/+}$ mice contain a GFP-encoding gene knocked into the exons 2–3 of the *S100a4* gene, resulting in a 50% reduction in S100a4 protein expression (*Xue et al., 2003*). To determine whether macrophages express S100a4 during healing, Csf1r-iCre; Rosa-Ai9; S100a4-GFP$^{promoter}$ mice were treated with Tamoxifen (Tmx; 100 mg/kg) to label Csf1r-lineage (Csf1r$^{Lin+}$) cells. For samples harvested on D3, mice were treated with Tmx on D0-2 post-surgery. Mice harvested at D14 were treated with Tmx on D0-2 post-surgery, and every other day thereafter until harvest. Cells actively expressing S100a4 were labeled green (S100a4-GFP$^{promoter+}$). For RAGE Antagonist Peptide (RAP) studies, C57BL/6J mice were treated with either 100 μg RAP or vehicle (0.5% bovine serum albumin in saline) via i.p. injection on D5-10 post-surgery.

### Murine model of tendon injury and repair

Male and female mice aged 10–12 weeks underwent complete transection and surgical repair of the flexor digitorum longus (FDL) tendon as previously described (*Ackerman and Loiselle, 2016*). Mice were monitored and given analgesics post-operatively as needed.

### RNA extraction and qPCR for in vivo studies

The tendon repair site was excised from the hind paw at D10 following injury, along with 1–2 mm of native tendon on either side. Three repairs were pooled, and RNA was extracted with TRIzol reagent (Life Technologies, Carlsbad CA). cDNA was generated with 500 ng of RNA using an iScript cDNA synthesis kit (BioRad, Hercules CA). Quantitative PCR was carried out with gene specific primers (*Table 1*), and expression normalized to *β-actin*. All experiments were done in biological triplicates and repeated twice (technical replicates).

### Assessment of gliding function and mechanical properties

Following sacrifice, the hindlimb was harvested at the knee. The medial side of the hindlimb was carefully dissected to free the FDL, and the proximal end was secured between two pieces of tape with cyanoacrylate. The distal tendon was loaded via the tape with weights ranging from 0 to 19 g, with digital images taken upon application of each weight. MTP Flexion Angle and Gliding Resistance were calculated as previously described (*Ackerman et al., 2017*; *Hasslund et al., 2008*; *Loiselle et al., 2009*), with lower MTP Flexion Angle and higher Gliding Resistance corresponding to restricted range of motion and impaired gliding function. Following gliding testing, the FDL was released from the tarsal tunnel, and the proximal end of the tendon and the digits were secured in opposing custom grips on an Instron 8841 uniaxial testing system (Instron Corporation, Norwood, MA). The tendon was loaded until failure at a rate of 30 mm/minute (*Hasslund et al., 2008*). Samples were excluded from analysis if the MTP Flexion Angle was less than 3° as dissection of samples below this threshold demonstrates consistent failure of the repair.

### Histology, Immunohistochemistry and Immunofluorescence

Following sacrifice, hind paws were dissected just above the ankle, and underwent routine processing for paraffin or frozen sectioning. Paraffin samples were fixed for 72 hr in 10% NBF, then decalcified for 2 weeks in 14% EDTA before processing. Three-micron sagittal sections were stained with Alcian Blue Hematoxylin/Orange G (ABH/OG) or picrosirius red stain (Polysciences Inc, Warrington PA). Frozen samples were fixed overnight, decalcified for 4 days, incubated in 30% sucrose (in PBS)

**Table 1.** qPCR Primer Sequences

| Gene | | Sequence (5'- > 3') | Reference |
|---|---|---|---|
| Actb | Fwd | AGATGTGCATCAGCAAGCAG | NM_007393.5 |
| | Rev | GCGCAAGTTAGGTTTTGTCA | |
| S100a4 | Fwd | AAGCTGAACAAGACAGAGCTCAAG | NM_011311.2 |
| | Rev | GTCCTTTTCCCCAGGAAGCTA | |
| Fn | Fwd | CGAGGTGACAGAGACCACAA | NM_001276413.1 |
| | Rev | CTGGAGTCAAGCCAGACACA | |
| Tnmd | Fwd | TGTACTGGATCAATCCCACTCT | NM_022322.2 |
| | Rev | GCTCATTCTGGTCAATCCCCT | |
| Scx | Fwd | TGGCCTCCAGCTACATTTCT | NM_198885.3 |
| | Rev | TGTCACGGTCTTTGCTGAAC | |
| Mkx | Fwd | CACCGTGACAACCCGTACC | NM_177595.4 |
| | Rev | GCACTAGCGTCATCTGCGAG | |
| Col1a1 | Fwd | GCTCCTCTTAGGGGCCACT | NM_007742.4 |
| | Rev | CCACGTCTCACCATTGGGG | |
| Col3a1 | Fwd | ACGTAGATGAATTGGGGATGCAG | NM_009930.2 |
| | Rev | GGGTTGGGGCAGTCTAGTG | |
| Acta2 | Fwd | GAGGCACCACTGAACCCTAA | NM_007392.3 |
| | Rev | CATCTCCAGAGTCCAGCACA | |
| iNOS | Fwd | CAGAGGACCCAGAGACAAGC | NM_001313921.1 |
| | Rev | TGCTGAAACATTTCCTGTGC | |
| TNFa | Fwd | AACTGTAAGCGGGGCAATCA | NM_013693.3 |
| | Rev | CCCCTTTCCTCCCAAACCAA | |
| Cd86 | Fwd | TCTCCACGGAAACAGCATCT | NM_019388.3 |
| | Rev | CTTACGGAAGCACCCATGAT | |
| Cd64 | Fwd | TCCTTCTGGAAAATACTGACC | NM_010186.5 |
| | Rev | GTTTGCTGTGGTTTGAGACC | |
| Cd206 | Fwd | CAGGTGTGGGCTCAGGTAGT | NM_008625.2 |
| | Rev | TGTGGTGAGCTGAAAGGTGA | |
| Arg1 | Fwd | AGGAACTGGCTGAAGTGGTTA | NM_007482.3 |
| | Rev | GATGAGAAAGGAAAGTGGCTGT | |
| IL1ra | Fwd | GCATCTTGCAGGGTCTTTTC | NM_001159562.1 |
| | Rev | GTGAGACGTTGGAAGGCAGT | |
| Cd163 | Fwd | TCCACACGTCCAGAACAGTC | NM_001170395.1 |
| | Rev | CCTTGGAAACAGAGACAGGC | |

DOI: https://doi.org/10.7554/eLife.45342.019

overnight, and embedded in Cryomatrix (#6769006, ThermoFisher, Waltham MA). Eight-micron sagittal sections on Cryofilm tape (Section-lab, Hiroshima, Japan) were cut on a Leica CM1860UV cryostat, and adhered to slides with 1% chitosan in 0.25% acetic acid.

Chromogen immunohistochemistry was performed on paraffin sections for S100a4 (1:20000, #197896, Abcam, Cambridge MA), with a rabbit polymer kit (#MP-7401, Vector Laboratories, Burlingame CA). Immunofluorescence was carried out with the following primary and secondary antibodies: RAGE (1:100, #sc-365154, Santa Cruz Biotechnology, Dallas TX), with goat anti-mouse AlexaFluor488 secondary (1:1000, #A11029, ThermoFisher, Waltham MA); F4/80 (1:500, #sc-26642, Santa Cruz Biotechnology, Dallas TX) with a donkey anti-rabbit AlexaFluor594 secondary (1:200, #711-585-152, Jackson ImmunoResearch, West Grove PA); iNOS (1:100, #Ab15323, Abcam), with a

Rhodamine-Red-X donkey anti-rabbit secondary (1:100, #711-296-152, Jackson ImmunoResearch); IL1Ra (1:10000, #Ab124962, Abcam), with a with a Rhodamine-Red-X donkey anti-rabbit secondary (1:100, #711-296-152, Jackson ImmunoResearch); GFP (1:5000, #ab6673, Abcam, Cambridge MA), with a donkey anti-goat 488 secondary (1:200, #705-546-147, Jackson ImmunoResearch, West Grove PA); RFP (1:500, #ab62341, Abcam, Cambridge MA), with a donkey anti-rabbit 647 secondary (1:200, #711-605-152, Jackson ImmunoResearch, West Grove PA); and α-SMA-Cy3 (1:250, #C6198 Sigma-Aldrich, St Louis MO). All slides were imaged with the Olympus slide scanner and processed with Olyvia software (Olympus, Waltham MA). Images were pseudo-colored using ImageJ software (v1.51j8, NIH). At least three animals were evaluated per genotype per time-point.

## Quantification of fluorescence

Slide scanned fluorescent images were analyzed with Visiopharm image analysis software v.6.7.0.2590 (Visiopharm, Hørsholm, Denmark) as previously described (*Ackerman et al., 2017*). Briefly, the percent area coverage of endogenous fluorescence or immunofluorescent staining was quantified in a semi-automated fashion using a threshold classifier for each fluorescent channel. Regions of interest were drawn to include only tendon and scar tissue. Manual correction excluded quantification of staining in blood vessels, auto-fluorescent sutures and debris smaller than 10 $\mu m^2$. Data are presented as the percent area of the region of interest that is positive for the appropriate fluorescent channel(s). Images were analyzed from 3 to 4 individual samples per genotype.

## In Vitro Studies

### Primary macrophage isolation

Bone marrow derived primary macrophages (BMDM) were grown from the bone marrow of C57Bl/6J, S100a4$^{GFP/+}$ and WT mice. Following sacrifice, femurs were flushed with ice-cold phosphate buffered saline (PBS, without Ca$^{2+}$/Mg$^{2+}$). The cell suspension was strained through a 70 $\mu m$ filter, resuspended in differentiation medium (*Weischenfeldt and Porse, 2008*) and plated at a concentration of $3 \times 10^6$ cells per 10 cm plate. At D7 of differentiation, primary macrophages were re-plated as needed for experimental use. All BMDM experiments were conducted in biological triplicates with 2–4 technical replicates.

### Macrophage Migration assay

Primary BMDMs were seeded at confluence in Oris 96-well plates (Platypus Technologies, Madison WI) with silicon stoppers inserted and incubated overnight. Plugs were removed, and cells washed once with PBS (with Ca$^{2+}$/Mg$^{2+}$) prior to addition of treatment. Vehicle (0.5% Bovine Serum Albumin in PBS) or recombinant S100a4 protein (S100a4-RP) was added to wells at 20, 50, 200, 500, and 1000 ng/mL in quadruplicate, and cells allowed to migrate for 24 hr. Following migration, cells were gently washed with PBS and stained with NucBlue Live ReadyProbe (Life Technologies, Carlsbad CA). A detection mask was affixed to the bottom of the plate to obscure cells that had not migrated, and the plate read on a Synergy two plate reader (Biotek, Winooski VT) at 360ex/480em, and fluorescence normalized to vehicle treated cells.

### Assessment of in vitro macrophage polarization

Primary BMDMs were seeded into 6-well plates at 85% confluence overnight. Cells were treated with either S100a4-RP (20–1000 ng/mL) or vehicle for 24 hr. Cells were lysed in Trizol (Life Technologies, Carlsbad CA), and RNA extracted using the RNeasy mini kit (Qiagen, Germantown MD). Quantitative PCR was performed with gene specific primers (*Table 1*) for markers of M1 (*iNOS*, *Tnfα*, *CD86*, *CD64*) and M2 (*CD206*, *Arg1*, *IL-1ra*, *CD163*) polarization, and data normalized expression in vehicle treated WT cells, and to *β-actin* expression.

### Primary tendon cell isolation

FDL tendons (n = 4 per genotype) were aseptically excised, pooled, and collagenase digested (0.075% collagenase I; #C6885, Sigma) in fibroblast grown medium-2 (FGM; #CC-3132, Lonza, Basel, Switzerland) for one hour at room temperature with stirring. The collagenase mixture was then filtered (70 $\mu M$ filter) and cells were pelleted, resuspended in FGM, and plated at 5,000 cells/cm$^2$ in collagen-coated plates (rat tail collagen type 1, 5 $\mu g$/cm$^2$; #354236, Corning, Tewksbury, MA). Cell

were maintained under standard culture conditions (37°C, 5% $CO_2$, 90% humidity), with complete media exchanges every other day. Upon reaching 70% confluence, cells were passaged using 0.05% trypsin-EDTA (#25300–054, Gibco, Waltham, MA).

## Tendon cell RNA isolation and qPCR

Total RNA was isolated from tendon cells at passage 1, by column purification (TRIzol Reagent; #15596026, Fisher Scientific; Direct-zol RNA Microprep kit, #R2061, Zymo Research) and converted to cDNA (qScript; #84034, Quantabio, Beverly, MA). Primers for murine genes of interest were designed (*Table 1*) (Primer Express, Applied Biosystems; *Table 1*), validated, and used for qPCR (PerfeCTa SYBR Green; #84069, Quantabio, CFX Connect Real-Time System; Bio-Rad). Data were normalized to β-actin (*Actb*) and expression in WT.

## Tendon cell proliferation assay

Tendon cells (passage 2) were plated onto collagen-coated black 96 well plates at 5,000 cells/cm$^2$ in FGM. Cell proliferation was evaluated every 24 hr for 4 days using the CellTiter-Glo Luminescent Cell Viability Assay (#PR-G7570, Fisher Scientific) in quadruplicate. Cell proliferation is shown relative to the average initial luminescent read for each genotype.

## Tendon cell scratch wound closure assay

Tendon cells (passage 2) were plated into collagen-coated 24 well plates in duplicate at 5,000 cells/cm$^2$ in FGM. At confluence, scratches were created in the cell monolayer using a p200 pipet tip. Monolayers were rinsed 3x with dPBS to remove debris and covered with fresh FGM containing 0.5% FBS. Triplicate images were taken at 0, 3, 5, 8, 12, and 24 hr to evaluate cell migration (ImageJ; National Institutes of Health).

## Statistical analyses and animal stratification

Statistically significant differences between genotypes or treatments in in vitro and in vivo studies were assessed by unequal variance un-paired t-test, with the following exceptions: The *S100a4* qPCR time-course was analyzed by one-way ANOVA, followed by Tukey's multiple comparisons test. The BMDM migration and polarization data, as well as the tenocyte proliferation and scratch wound assay were analyzed by two-way ANOVA with Sidak's multiple comparison test. All analyses were conducted using GraphPad Prism software (v8.0.0, La Jolla CA). Data are presented as mean ± SD. p values ≤ 0.05 were considered significant, with the following conventions: *=p ≤ 0.05, **=p ≤ 0.01, and ***=p ≤ 0.001.: Mice were randomly allocated to specific experimental outcome metrics prior to surgery. Analysis of subjective quantitative data (MTP Flexion Angle, Gliding Resistance) were done in a blinded manner. Outlier data points for tested for using GraphPad Prism software using the ROUT method, and the Q value set at 1%, however no outliers were identified in any quantitative data sets.

# Acknowledgements

We would like to thank the Histology, Biochemistry and Molecular Imaging (HBMI) and the Biomechanics, Biomaterials and Multimodal Tissue Imaging (BBMTI) Cores for technical assistance. This work was supported in part by NIH/NIAMS K01AR068386 and R01AR073169 (to AEL). The HBMI and BBMTI Cores are supported by NIH/NIAMS P30AR069655.

# Additional information

## Funding

| Funder | Grant reference number | Author |
|---|---|---|
| National Institute of Arthritis and Musculoskeletal and Skin Diseases | K01AR068386 | Alayna E Loiselle |

| National Institute of Arthritis and Musculoskeletal and Skin Diseases | R01AR073169 | Alayna E Loiselle |
|---|---|---|

The funders had no role in study design, data collection and interpretation, or the decision to submit the work for publication.

## Author contributions
Jessica E Ackerman, Conceptualization, Data curation, Formal analysis, Writing—original draft, Writing—review and editing; Anne EC Nichols, Data curation, Formal analysis; Valentina Studentsova, Katherine T Best, Data curation, Formal analysis, Writing—review and editing; Emma Knapp, Data curation, Writing—review and editing; Alayna E Loiselle, Conceptualization, Formal analysis, Funding acquisition, Writing—original draft, Writing—review and editing

## Author ORCIDs
Alayna E Loiselle (iD) https://orcid.org/0000-0002-7548-6653

## Ethics
Animal experimentation: This study was performed in strict accordance with the recommendations in the Guide for the Care and Use of Laboratory Animals of the National Institutes of Health. All animal studies were approved by the University of Rochester Committee for Animal Resources (Protocol 2014-004).

## Decision letter and Author response
Decision letter https://doi.org/10.7554/eLife.45342.022
Author response https://doi.org/10.7554/eLife.45342.023

## Additional files

### Supplementary files
• Transparent reporting form
DOI: https://doi.org/10.7554/eLife.45342.020

### Data availability
All data generated or analyzed during this study are included in the manuscript and supporting files.

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
