## [Decision Letter]

Thank you for submitting your article "Cell non-autonomous functions of S100a4 drive fibrotic tendon healing" for consideration by *eLife*. Your article has been reviewed by three peer reviewers, including Clifford J Rosen as the Reviewing Editor and Reviewer #1, and the evaluation has been overseen by Harry Dietz as the Senior Editor. The following individuals involved in review of your submission have agreed to reveal their identity: Stavros Thomopoulos (Reviewer #3).

The reviewers have discussed the reviews with one another and the Reviewing Editor has drafted this decision to help you prepare a revised submission.

Summary:

All three reviewers were positively inclined and felt the data are novel and potentially important. You have shown that S100a4 drives fibrotic tendon healing, possibly through a cell non-autonomous process; S100a4 haploinsufficiency promoted regenerative tendon healing and inhibition of S100a4 via antagonism of its putative receptor, RAGE, decreased scar formation. Also knock-down of S100a4 decreased myofibroblast and macrophage content at the site of injury, with both cell populations being key drivers of fibrotic progression. Finally, S100a4-lineage cells were shown to convert into α-SMA^+^ myofibroblasts. On the other hand, one of the major concerns that the reviewers had was your conclusion that these changes were all due to cell non- autonomous actions. This needs to be addressed more completely since S100a4 is expressed in the tendon. Moreover, the issue of how the macrophages are contributing to the phenotype should be explored. Other important comments are noted below but these two issues require detailed characterization.

Reviewer #1:

In the present study the authors showed that S100a4 drives fibrotic tendon healing through a cell non-autonomous process; S100a4 haploinsufficiency promoted regenerative tendon healing and inhibition of S100a4 via antagonism of its putative receptor, RAGE, decreased scar formation. Also knock-down of S100a4 decreased myofibroblast and macrophage content at the site of injury, with both cell populations being key drivers of fibrotic progression. Finally, S100a4-lineage cells were shown to convert into α-SMA^+^ myofibroblasts.

Overall, the manuscript provides some new insight into the role of early, S100a4 progenitors in promoting fibrous healing, a major clinical problem. In essence, the authors have shown the cell non-autonomous actions via the global knockout on tendon healing, but we are left not understanding the relative contribution of the cell autonomous actions, nor is the mechanism of this multifunctional protein during injury clearly elucidated. The lineage tracing supporting a transition from S100a4 to α-SMA are consistent and are important results. But several questions arise from the data presented:

1) Were there any sex differences in the haploinsufficiency mice relative the injury phenotype? to circulating levels of S100a4.

2) Can the authors provide a quantitation of the increase in highly aligned, mature collagen fibers in the haploinsufficient model.

3) One essential element that is missing in balancing cell autonomous vs cell non-autonomous actions. Is a macrophage specific condition deletion or a tenocyte conditional mouse available? Have the authors considered how else to distinguish tenocyte vs macrophage derived S100a4 effects on tendon healing?

4) Does S100a4 bind only to the RAGE receptor or are there other receptors? Is RAP specific for the RAGE receptor or does it have other ligands? Is the RAGE null mouse available and would that mouse show enhanced healing after injury?

5) The S100a4-TK experiment demonstrated a marked reduction in strength post injury? This would imply a dose dependent difference in the impact on healing of S100a4 and complicates the interpretation. Thus, there must be a complex role for S100a4 in tendon healing. Are there any in vitro data to support the kind of relationship that is U-shaped? Is it conceivable that the temporal deletion using the TK mouse might have other cell non-autonomous effects from GCV- TK? Is the timing of the temporal deletion critical for the S100a4 effects?

Reviewer #2:

In this manuscript Ackerman et al. show that S100a4 haploinsufficiency improves tendon healing and antagonism of its putative receptor decreases scar formation. The ability to show improved tendon healing properties in a partial loss of function model are very exciting findings. The authors present possible mechanisms for how S100a4 may be acting during healing, including regulating macrophage phenotypes, matrix gene expression, and formation of SMA^+^ myofibroblasts. I think the paper would be of interest to the broad readership at *eLife*, but there are some issues that should first be addressed.

The main concern is in regards to the authors' conclusion that they have demonstrated that S100a4 promotes scar-mediated healing via 'cell non-autonomous mechanisms'. It would help if the authors could better explain how their data supports this conclusion. The authors show that S100a4-active and lineage cells are found throughout the tendon before and during healing. It also appears that the cells that are expressing S100a4 are co-expressing RAGE (Figure 5A). In addition, the use of haploinsufficient S100a4 or RAP injection to disrupt S100a4 gene/pathway function does so throughout all tissues in the animal, making it difficult to interpret cell autonomous or cell non-autonomous mechanisms. Cell or temporal specific loss of function studies and demonstrating that S100a4 is signaling through RAGE seem necessary to make such conclusions.

Is there a difference in tendon properties in the S100a4 hets compared to wild types prior to injury? It appears that many cells in the tendon are S100a4^GFP+^ prior to injury, and it is unclear if S100a4 haploinsufficiency results in tendons that are different prior to injury (Figure 2D-G). As this haploinsufficient model is not tissue or temporally specific, it would help to distinguish injury-specific roles of S100a4 vs a general role in the tendon.

Please indicate the stages of healing that are shown in Figure 3. In Figure 3F, it appears that there are more vessels with strong SMA labeling in the S100a4 hets. Do the S100a4 hets have increased vasculature in the healing area?

There are a few figures where cells are described as 'decreased', 'few' or 'most' (subsection “S100a4 modulates macrophage function”, Discussion section). It would help if there was some sort of quantification of this data. Specifically- the reduction in macrophages for Figure 4A and a quantification of the SMA co-expressing cells populations.

The authors show exogenous addition of S100a4 is sufficient to alter macrophage gene expression, but their model of improved tendon healing is with S100a4 haploinsufficiency. Does S100a4 haploinsufficiency alter macrophage phenotypes? Previous studies (Dulyaniova et al., 2018 and Li et al., 2010) showing S100a4 is necessary for macrophage chemotaxis use S100a4^-/-^ null mutants. It would better support their model if they showed that S100a4 het macrophages have abnormal migration or polarization.

There appear to be increased S100a4 cells near the sutures in Figure 6C. Shouldn't these cells be depleted or reduced in the S100a4-TK model?

The authors conclude that RAP-mediated inhibition of S100a4-RAGE activity is insufficient to improve mechanical properties (in contrast with S100a4 hets) and that S100a4 may act independently of RAGE (Discussion section). It is also possible that a constitutive haploinsufficiency (loss throughout development and adulthood in all tissues) may not recapitulate an acute pathway loss during adult healing. The possibility that S100a4 haploinsufficiency causes earlier defects or changes in other tissues that could indirectly cause this phenotype should also be considered as there are differences in these loss of function models.

Reviewer #3:

The authors present an excellent study examining the role of S1004a in tendon healing. They use multiple mouse models, including reporters, cell ablation, and haploinsufficiency to demonstrate S1004a involvement in tendon healing. Furthermore, using genetic and pharmacological approaches, they show the therapeutic potential for targeting S1004a in tendon healing.

1) The authors should expand their Discussion section on the limitations of the flexor tendon injury and repair model. This model does not approximate human flexor tendon injury and repair, as rehabilitation cannot be implemented. They should also modify their Introduction, to describe tendon healing in general, and remove the specific reference to flexor tendon repairs. Nonetheless, their approach is a good basic science model model for studying tendon healing.

2) The authors are commended for measuring gliding function and tensile properties. However, they do not include normalized (material) properties such as failure stress, modulus, and toughness/energy. These properties require the measurement of cross sectional area, which would also serve as a valuable measure of scar formation.

3) Data should be presented as mean +/- standard deviation, which I believe is a more valuable measure of the variability in the measures.

4) Figure 1D shows that, at D3, S100a4 is not expressed in tendon, but it is expressed again by D7. Is this an artifact of the imaging, or does the expression pattern in the tendon really shift dramatically over the course of a few days?

5) The authors should add quantification of the patterns shown in B and D, so that the percentage of S100a4-lineage vs. S100a4-active cells can be determined in the tendon and scar.

6) The results for S100a4^GFP/+^ are impressive. The authors validated a ~50% decrease in s100a4 at the gene and protein levels. They found increased range of motion and increase failure load. Typically, tendon treatment approaches that suppress fibrosis lead to increased motion but also to decreased strength. These results imply regeneration, not just decreased ECM production. However, the authors did not report material properties; decreased scar and increased failure load implies even better improvements in failure stress and modulus. Furthermore, 14d of healing is a relatively short time frame for collagen production and remodeling, and a longer healing timepoint would be valuable. These points should be discussed by the authors.

7) In Figure 4, the authors should add sections stained for M1 vs. M2 phenotype. They can also look for M1 vs. M2 markers using qPCR. These assessments from in vivo samples would reinforce the relevance of the very strong in vitro studies.

8) The RAGE antagonist experiments in Figure 5 nicely show the therapeutic potential of targeting S100a4.

9) The results in Figure 6 and Figure 7 are more typical of attempts to shut down fibrosis: improved range of motion but decreased strength. A puzzling result is that complete ablation throughout the healing period (Figure 7) led to decreased ROM and decreased strength. This isn't logical and should be discussed by the authors.

10) The schematic in Figure 8 is speculative. The authors should perform the experiments with an inducible Cre and in using tendon-specific targeting to more accurately trace the source and lineage of the cells. Alternatively, in situ hybridization can be performed to track expression patterns over time.

11) The authors should discuss the limitations that their mouse models are neither tendon specific nor inducible.

---

## [Author Response]

Summary:All three reviewers were positively inclined and felt the data are novel and potentially important. You have shown that S100a4 drives fibrotic tendon healing, possibly through a cell non-autonomous process; S100a4 haploinsufficiency promoted regenerative tendon healing and inhibition of S100a4 via antagonism of its putative receptor, RAGE, decreased scar formation. Also knock-down of S100a4 decreased myofibroblast and macrophage content at the site of injury, with both cell populations being key drivers of fibrotic progression. Finally, S100a4-lineage cells were shown to convert into α-SMA^+^ myofibroblasts. On the other hand, one of the major concerns that the reviewers had was your conclusion that these changes were all due to cell non- autonomous actions. This needs to be addressed more completely since S100a4 is expressed in the tendon. Moreover, the issue of how the macrophages are contributing to the phenotype should be explored. Other important comments are noted below but these two issues require detailed characterization.Reviewer #1:In the present study the authors showed that S100a4 drives fibrotic tendon healing through a cell non-autonomous process; S100a4 haploinsufficiency promoted regenerative tendon healing and inhibition of S100a4 via antagonism of its putative receptor, RAGE, decreased scar formation. Also knock-down of S100a4 decreased myofibroblast and macrophage content at the site of injury, with both cell populations being key drivers of fibrotic progression. Finally, S100a4-lineage cells were shown to convert into α-SMA^+^ myofibroblasts.Overall, the manuscript provides some new insight into the role of early, S100a4 progenitors in promoting fibrous healing, a major clinical problem. In essence, the authors have shown the cell non-autonomous actions via the global knockout on tendon healing, but we are left not understanding the relative contribution of the cell autonomous actions, nor is the mechanism of this multifunctional protein during injury clearly elucidated. The lineage tracing supporting a transition from S100a4 to α-SMA are consistent and are important results. But several questions arise from the data presented:1) Were there any sex differences in the haploinsufficiency mice relative the injury phenotype? to circulating levels of S100a4.

The healing phenotype trends observed when male and female data are analyzed together were consistent when each sex was analyzed separately. For example, gliding resistance was significantly reduced in male and female S100a4^GFP/+^ repairs, relative to sex-matched WT repairs. MTP flexion angle and max load were increased in male and female S100a4-^GFP/+^ repairs, relative to sex-matched WT repairs. These differences were not statistically significant, likely due to the limited power of the smaller data sets. While there is no clear evidence of sexual dimorphism with respect to S100a4 levels or expression in the literature, S100a4^-/-^ mice are born with an abnormal sex distribution (fewer females). In addition, Erlandsson et al., demonstrated that female S100a4^-/-^ mice were resistant to ovariectomy-induced bone loss, relative to WT (Erlandsson et al., 2013). Moreover, Dempsie et al., demonstrate that overexpression of S100a4 in females leads to increased pulmonary artery hypertension (PAH) lesions, while S100a4 overexpression in males does not have an effect and they suggest that the higher incidence of pulmonary artery hypertension that is observed in females is due to elevated levels of S100a4 (Dempsie et al., 2011).

2) Can the authors provide a quantitation of the increase in highly aligned, mature collagen fibers in the haploinsufficient model.

We attempted to use Second Harmonic Generation multiphoton imaging to analyze collagen alignment, however this approach was unsuccessful due to technical limitation, likely related to sample thickness. Since we were not able to provide quantitative data to support this conclusion, we have revised the Results section as appropriate.

3) One essential element that is missing in balancing cell autonomous vs cell non-autonomous actions. Is a macrophage specific condition deletion or a tenocyte conditional mouse available? Have the authors considered how else to distinguish tenocyte vs macrophage derived S100a4 effects on tendon healing?

This is a very good point, and something we are interested in addressing. We are currently generating both macrophage and tendon conditional S100a4 knockout animals, but these studies are just beginning. To begin to identify the specific cells that express S100a4 during tendon healing we have crossed S100a4-GFP^promoter+^ mice to Csf1r-CreER; Rosa-Ai9 mice, which allows us to determine whether macrophage lineage cells actively express S100a4 during healing. These data, which are presented in new Figure 4—figure supplement 1 demonstrate that many macrophages express S100a4 during early tendon healing, and support the future use of macrophage conditional S100a4 deletion. Moreover, there is also a population of S100a4^+^ cells that are not Csf1r-lineage, supporting the future use of additional Cre drivers to delineate the relative contributions of discrete populations of S100a4^+^ cells.

To address the cell autonomous vs. cell non-autonomous function of S100a4 we isolated primary bone marrow derived macrophages (BMDMs) and primary tendon cells from WT and S100a4^GFP/+^ mice. No changes in BMDM migration in vehicle treated BMDM were observed between WT and S100a4^GFP/+^ cells, and there was a consistent enhancement in migration in response to exogenous S100a4 in both genotypes (Figure 4—figure supplement 2B). Thus, suggesting that S100a4 haploinsufficiency does not alter intrinsic macrophage function and that the macrophage response to S100a4 is largely cell non-autonomous. In addition, no changes in polarization were observed between WT and S100a4^GFP/+^ BMDMs (Figure 4—figure supplement 2E and F). In primary tendon cells, no changes in expression of tenogenic or matrix related genes were observed between WT and S100a4^GFP/+^ cells (Figure 4—figure supplement 3A and B). In addition, S100a4 haploinsufficiency did not alter tendon cell proliferation (Figure 4—figure supplement 3C). However, migration, as assessed by scratch wound closure, was significantly increased in S00a4^GFP/+^ tendon cells (Figure 4—figure supplement 3D), suggesting that S100a4 has both cell autonomous and cell non-autonomous functions in tendon cells.

4) Does S100a4 bind only to the RAGE receptor or are there other receptors? Is RAP specific for the RAGE receptor or does it have other ligands? Is the RAGE null mouse available and would that mouse show enhanced healing after injury?

In addition to RAGE, there is evidence that S100a4 can interact with TLR4 (Björk et al.), EGFR ligands (Klingelhofer et al., 2009), Heparan Sulfate Proteoglycans (Kiryushko et al., 2006), and cell surface Annexin 2 (Semov et al., 2005). While the role of these molecules in tendon healing is not clear, it will be important to define the RAGE independent mechanisms of S100a4 in future studies, and we have discussed this in the revised manuscript (Discussion section).

RAP is specific for blocking ligand binding to RAGE, however, RAP also prevents other RAGE ligands, such as HMGB-1 and S100P from binding. Thus, we believe that the differences in healing phenotypes observed between S100a4^GFP/+^ and RAP treated animals may be due to a combination of S100a4 signaling through receptors other than RAGE, and inhibition of RAGE signaling via ligands other than S100a4. We have included these as limitations in the Discussion section.

We have recently received RAGE^-/-^ mice and anticipate that these mice will heal in a manner similar to RAP treated animals, however, these mice will also clarify the effects of sustained RAGE inhibition, relative to the delayed inhibition that occurs with RAP.

5) The S100a4-TK experiment demonstrated a marked reduction in strength post injury? This would imply a dose dependent difference in the impact on healing of S100a4 and complicates the interpretation. Thus, there must be a complex role for S100a4 in tendon healing. Are there any in vitro data to support the kind of relationship that is U-shaped? Is it conceivable that the temporal deletion using the TK mouse might have other cell non-autonomous effects from GCV- TK? Is the timing of the temporal deletion critical for the S100a4 effects?

This raises two important points regarding S100a4: i) dose-dependent effects and, ii) time-dependent effects.

In terms of timing, it is clear that the timing cell depletion alters the healing response (Figure 6 and Figure 7). However, we have not addressed the effects of inducible or delayed S100a4 haploinsufficiency. Moreover, it is likely that modifying the timing of RAP treatment to inhibit S100a4-RAGE signaling would alter the healing response, and based on the effects of S100a4^+^ cell depletion during the acute inflammatory phase of healing (S100a4-TK[D1-14]; Figure 7), we would hypothesize that RAP treatment at this time would also be detrimental. Future studies using an inducible Cre driver and S100a4^flox^ mice will provide important clarity on the time-dependent functions of S100a4 during healing, which we have addressed in the Discussion section.

In terms of a dose-dependent effect on healing, this is more complicated to address particularly in the S100a4-TK mice, since as the reviewer notes there are likely to be non-autonomous effects on other cells due to the increase in apoptotic cells in the healing environment. With respect to in vitro data, the data in Figure 4—figure supplement 2A modestly suggest a potential U-shaped response to exogenous S100a4, as significant increases in migration are observed in WT bone marrow derived macrophages in response to 50, 200, and 100ng/mL recombinant S100a4, while no increases are observed with 500ng/mL treatment. However, these data are not sufficient to clearly demonstrate dose-dependent effects of S100a4.

Reviewer #2:In this manuscript Ackerman et al. show that S100a4 haploinsufficiency improves tendon healing and antagonism of its putative receptor decreases scar formation. The ability to show improved tendon healing properties in a partial loss of function model are very exciting findings. The authors present possible mechanisms for how S100a4 may be acting during healing, including regulating macrophage phenotypes, matrix gene expression, and formation of SMA^+^ myofibroblasts. I think the paper would be of interest to the broad readership at eLife, but there are some issues that should first be addressed.The main concern is in regards to the authors' conclusion that they have demonstrated that S100a4 promotes scar-mediated healing via 'cell non-autonomous mechanisms'. It would help if the authors could better explain how their data supports this conclusion. The authors show that S100a4-active and lineage cells are found throughout the tendon before and during healing. It also appears that the cells that are expressing S100a4 are co-expressing RAGE (Figure 5A). In addition, the use of haploinsufficient S100a4 or RAP injection to disrupt S100a4 gene/pathway function does so throughout all tissues in the animal, making it difficult to interpret cell autonomous or cell non-autonomous mechanisms. Cell or temporal specific loss of function studies and demonstrating that S100a4 is signaling through RAGE seem necessary to make such conclusions.Is there a difference in tendon properties in the S100a4 hets compared to wild types prior to injury? It appears that many cells in the tendon are S100a4^GFP+^ prior to injury, and it is unclear if S100a4 haploinsufficiency results in tendons that are different prior to injury (Figure 2D-G). As this haploinsufficient model is not tissue or temporally specific, it would help to distinguish injury-specific roles of S100a4 vs a general role in the tendon.

This is very good point. To address this, we evaluated the gliding and biomechanical properties of uninjured contralateral FDL tendons. No significant differences in MTP Flexion Angle, Gliding Resistance, max load at failure, or stiffness were observed between genotypes. This data is presented in Figure 2—figure supplement 1.

Please indicate the stages of healing that are shown in Figure 3. In Figure 3F, it appears that there are more vessels with strong SMA labeling in the S100a4 hets. Do the S100a4 hets have increased vasculature in the healing area?

The data in Figure 3 were collected at 14 days post-surgery, which corresponds to the proliferative phase of healing. We have updated both the text and figure legend to include this information.

In terms of increased vasculature in S100a4^GFP/+^ repairs, we examined n=4-5 specimens per genotype and observed a relatively consistent level of vascularity in the healing area between WT and S100a4^GFP/+^ repairs. We agree that there is more pronounced α-SMA staining in the vessels of S10a04^GFP/+^ repairs in Figure 3F, though this may also be due to the plane of section through the vessels. Interestingly, there is evidence of synergistic effects for S100a4 and VEGF (via RAGE signaling) to promote endothelial cell migration and enhanced vascularity (Grum-Schwensen et al., 2005; Hernandez et al., 2013; Ochiya, Takenaga and Endo, 2014), which would actually suggest the hypothesis of decreased vasculature in S100a4^GFP/+^ repairs.

There are a few figures where cells are described as 'decreased', 'few' or 'most' (subsection “S100a4 modulates macrophage function”, Discussion section). It would help if there was some sort of quantification of this data. Specifically- the reduction in macrophages for Figure 4A and a quantification of the SMA co-expressing cells populations.

We have now included quantification of the S100a4^Lin+^ population and S100a4-GFP^promoter+^ population over time (new Figure 1C and F), F4/80, iNOS and IL1ra expression in WT and S100a4^GFP/+^ repairs (new Figure 4A-D), and the S100a4^Lin+^; α-SMA^+^ and S100a4-GFP^promoter+^; α-SMA^+^ populations (new Figure 8C). We have also revised the Results section and Discussion section to reflect the conclusions that can be drawn from this quantitative data and have updated the Materials and methods section.

The authors show exogenous addition of S100a4 is sufficient to alter macrophage gene expression, but their model of improved tendon healing is with S100a4 haploinsufficiency. Does S100a4 haploinsufficiency alter macrophage phenotypes? Previous studies (Dulyaniova et al., 2018 and Li et al., 2010) showing S100a4 is necessary for macrophage chemotaxis use S100a4^-/-^ null mutants. It would better support their model if they showed that S100a4 het macrophages have abnormal migration or polarization.

This is a very good point. To address this, we isolated primary bone marrow derived macrophages (BMDM) from S100a4^GFP/+^ and WT mice. These data demonstrate that S100a4 haploinsufficiency does not alter macrophage migration, nor does it impair the pro-migratory effects of exogenous S100a4 on macrophages (Figure 4—figure supplement 2B). These data are in contrast to the migration phenotype observed with complete deletion of S100a4, suggesting that there may be an S100a4 expression threshold below which defects in migration occur. In addition, it is unknown how S100a4^-/-^ macrophages respond to exogenous S100a4 treatment, which would further clarify the roles of cell autonomous vs. non-autonomous S100a4 in macrophages. These data are now further discussed in the Discussion. In addition, no changes in polarization were observed between WT and S100a4^GFP/+^ BMDMs at baseline, or in response to exogenous S100a4 (Figure 4—figure supplement 2E and F).

There appear to be increased S100a4 cells near the sutures in Figure 6C. Shouldn't these cells be depleted or reduced in the S100a4-TK model?

We examined n=4 samples of S100a4-TK (D5-10) mice at D14 and consistently observed a persistent S100a4+ population for which the image in Figure 6C is representative. Lack of depletion of these cells is either due to incomplete depletion of proliferating S100a4-TK cells with the GCV regimen (GCV has a very short half-life so is administered every 12hrs, however even this regimen is not likely to achieve 100% depletion), or because these cells were not actively proliferating and/or expressing S100a4 during GCV treatment. That is, since GCV treatment was ended at D10 and the histology and gliding/ mechanics samples were harvested at D14 there is likely a population of non-depleted cells that express S100a4 between D10-14. We have revised the Results section, Materials and methods section and figure legends to clarify the timing of histological analysis of S100a4-TK (D5-10) mice.

The authors conclude that RAP-mediated inhibition of S100a4-RAGE activity is insufficient to improve mechanical properties (in contrast with S100a4 hets) and that S100a4 may act independently of RAGE (Discussion section). It is also possible that a constitutive haploinsufficiency (loss throughout development and adulthood in all tissues) may not recapitulate an acute pathway loss during adult healing. The possibility that S100a4 haploinsufficiency causes earlier defects or changes in other tissues that could indirectly cause this phenotype should also be considered as there are differences in these loss of function models.

This is a good point. As noted above, we now provide data that S100a4 haploinsufficiency does not significantly alter gliding function or mechanical properties of un-injured adult flexor tendons (Figure 2—figure supplement 1), suggesting that S100a4 haploinsufficiency, and therefore diminished S100a4-RAGE signaling does not disrupt tendon homeostasis. However, in addition to acknowledging that S100a4 may act independently of RAGE, we now also include text to reflect the possibility that S100a4 haploinsufficiency in other cells or tissues may impact the tendon healing phenotype in S100a4^GFP/+^ repairs (Discussion section).

Reviewer #3:The authors present an excellent study examining the role of S1004a in tendon healing. They use multiple mouse models, including reporters, cell ablation, and haploinsufficiency to demonstrate S1004a involvement in tendon healing. Furthermore, using genetic and pharmacological approaches, they show the therapeutic potential for targeting S1004a in tendon healing.1) The authors should expand their Discussion section on the limitations of the flexor tendon injury and repair model. This model does not approximate human flexor tendon injury and repair, as rehabilitation cannot be implemented. They should also modify their Introduction, to describe tendon healing in general, and remove the specific reference to flexor tendon repairs. Nonetheless, their approach is a good basic science model model for studying tendon healing.

We have acknowledged the limitations of this model of healing in revised manuscript (Discussion section), including the importance of physical therapy to improving the healing process. We have also revised the introduction to reflect the general process of tendon healing and have added references to support the complications of healing in other tendons rather than focusing only on the flexor tendon (Introduction).

2) The authors are commended for measuring gliding function and tensile properties. However, they do not include normalized (material) properties such as failure stress, modulus, and toughness/energy. These properties require the measurement of cross sectional area, which would also serve as a valuable measure of scar formation.

Thank you for this comment, it is an important point. Given that the measures of gliding function and tensile properties are conducted with the tendon in situ to maintain and account for scar tissue adhesions between the tendon and surrounding tissues, it has not been possible to measure cross-sectional area and therefore to calculate material properties. However, we are currently developing a non-invasive ultrasound approach that will facilitate measurement of cross-sectional area and subsequent calculation of material properties in future studies.

3) Data should be presented as mean +/- standard deviation, which I believe is a more valuable measure of the variability in the measures.

Quantitative data are now presented +/- Standard deviation.

4) Figure 1D shows that, at D3, S100a4 is not expressed in tendon, but it is expressed again by D7. Is this an artifact of the imaging, or does the expression pattern in the tendon really shift dramatically over the course of a few days?

There are a small number of S100a4^+^ cells in the tendon stubs at D3, however this low cellularity is consistent both between S100a4-GFP^promoter^ samples from this experiment and general histological assessment of D3 in this model. This finding is consistent with a transient decrease in the cellularity of the tendon ends during early healing, as Wu et al., have shown that apoptosis of tenocytes peaks at D3 following injury (Wu et al., 2010), followed by migration of tenocytes from tendon tissue away from the repair to the tendon ends (Sharma and Maffulli, 2006), as well as the likely influx of extrinsic cells.

5) The authors should add quantification of the patterns shown in B and D, so that the percentage of S100a4-lineage vs. S100a4-active cells can be determined in the tendon and scar.

We have now provided quantification of these data, which are presented in new Figure 1C and F. Interestingly, quantification of S100a4GFP^promoter+^ cells demonstrates peak percent S100a4^+^ area at D14 post-surgery. Given that we quantified% (+) area it is not possible to directly compare Lin^+^ vs. Promoter^+^ cells, however, we are currently generating S100a4^Lin+^; S100a4-GFP^promoter+^ mice to understand the relationship between these populations in future work.

6) The results for S100a4^GFP/+^ are impressive. The authors validated a ~50% decrease in s100a4 at the gene and protein levels. They found increased range of motion and increase failure load. Typically, tendon treatment approaches that suppress fibrosis lead to increased motion but also to decreased strength. These results imply regeneration, not just decreased ECM production. However, the authors did not report material properties; decreased scar and increased failure load implies even better improvements in failure stress and modulus. Furthermore, 14d of healing is a relatively short time frame for collagen production and remodeling, and a longer healing timepoint would be valuable. These points should be discussed by the authors.

This is a good point. As noted above we are developing an ultrasound imaging approach to measure cross-sectional area, which will allow calculation of material properties in future studies. We have included lack of material property characterization as a limitation of this study (Discussion section). In addition, we have also discussed the limitation of assessing healing only at 14 days post-surgery (Discussion section).

7) In Figure 4, the authors should add sections stained for M1 vs. M2 phenotype. They can also look for M1 vs. M2 markers using qPCR. These assessments from in vivo samples would reinforce the relevance of the very strong in vitro studies.

We have now included staining for M1 (iNOS) and M2 (IL1ra) macrophage markers (new Figure 4B and C), which demonstrates a reduction in M1 macrophages, and no effect on M2 macrophages in S100a4^GFP/+^ repairs relative to WT, supporting a less inflammatory environment in S100a4^GFP/+^ repairs.

8) The RAGE antagonist experiments in Figure 5 nicely show the therapeutic potential of targeting S100a4.

Thank you.

9) The results in Figure 6 and Figure 7 are more typical of attempts to shut down fibrosis: improved range of motion but decreased strength. A puzzling result is that complete ablation throughout the healing period (Figure 7) led to decreased ROM and decreased strength. This isn't logical and should be discussed by the authors.

We agree that the well-characterized response to disrupting the fibrotic process, typically via inhibition of the acute inflammatory phase, is improved ROM and decreased mechanics. While it is not entirely clear why depletion of S100a4-TK^+^ cells from D1-14 does not improve ROM (Figure 7), these data may suggest that it is actually the depletion of the cells themselves that is driving this unexpected decrease in ROM, rather than the decrease in S100a4 expression. As we begin to identify the different cell populations that express S100a4 during tendon healing, we will not only be able to define the discrete roles of S100a4 production from these cells, but it will also help clarify why sustained depletion of S100a4^+^ cells decreases ROM, and which cell populations are particularly involved in scar formation and remodeling.

10) The schematic in Figure 8 is speculative. The authors should perform the experiments with an inducible Cre and in using tendon-specific targeting to more accurately trace the source and lineage of the cells. Alternatively, in situ hybridization can be performed to track expression patterns over time.

We have revised the schematic to reflect the diversity of S100a4^+^ cells, including macrophages and tendon cells, and acknowledge in both the figure legend and the text that there are multiple S100a4-expressing cell types, including those yet to be identified. We also clarify that this a proposed mechanism of S100a4 during pro-fibrotic tendon healing, rather than conclusions drawn from the data. We agree that S100a4 conditional deletion in tendon cells will be important, and we plan to conduct these studies, unfortunately that work cannot be completed in a reasonable timeframe and are outside the scope of the present study. We have discussed these limitations in the revised manuscript (Discussion section).

11) The authors should discuss the limitations that their mouse models are neither tendon specific nor inducible.

We have included a discussion of these limitations within the text of the manuscript (Discussion section).